# Artificial intelligence driven tumor risk stratification from single-cell transcriptomics using phenotype algebra

**Namrata Bhattacharya[1,2,3], Anja Rockstroh[1,3], Sanket Suhas Deshpande[4], Sam Koshy Thomas[5], Anunay Yadav[2], Chitrita Goswami[2], Smriti Chawla[6], Pierre Solomon[7], Cynthia Fourgeux[7], Gaurav Ahuja[4,8], Brett Hollier[1,3], Himanshu Kumar[9], Antoine Roquilly[7], Jeremie Poschmann[7], Melanie Lehman[1,10], Colleen C Nelson[1,3]\*, Debarka Sengupta[2,4,8]\***

[1]Australian Prostate Cancer Research Centre-Queensland, Faculty of Health, School of Biomedical Sciences, Centre for Genomics and Personalised Health, Queensland University of Technology, Brisbane, Australia; [2]Department of Computer Science and Engineering, Indraprastha Institute of Information Technology-Delhi (IIIT-Delhi), Okhla, Phase III, New Delhi, India; [3]Translational Research Institute, Princess Alexandra Hospital, Woolloongabba, Australia; [4]Department of Computational Biology, Indraprastha Institute of Information Technology-Delhi (IIIT-Delhi), Okhla, Phase III, New Delhi, India; [5]School of Mathematical Sciences, The University of Adelaide, Adelaide, Australia; [6]Center for Computational Biomedicine, Harvard Medical School, Boston, United States; [7]Nantes Université, CHU Nantes, INSERM, Center for Research in Transplantation and Translational Immunology, UMR, Nantes, France; [8]Centre for Artificial Intelligence, Indraprastha Institute of Information Technology-Delhi (IIIT-Delhi), Okhla, Phase III, New Delhi, India; [9]Laboratory of Immunology and Infectious Disease Biology, Department of Biological Sciences, Indian Institute of Science Education and Research (IISER), Bhopal, India; [10]Vancouver Prostate Centre, Department of Urologic Sciences, University of British Columbia, Vancouver, Canada

**\*For correspondence:**
colleen.nelson@qut.edu.au
(CCN);
debarka@iiitd.ac.in (DS)

## eLife Assessment

This manuscript presents an **important** contribution to the field of single-cell transcriptomic analysis in cancer by introducing a novel computational framework-SCellBOW-which applies embedding techniques from natural language processing to model phenotypic heterogeneity in tumors. The revised version includes new validation experiments and significant clarifications that provide **convincing** evidence for the method's utility. The authors have benchmarked SCellBOW across diverse datasets, including glioblastoma, breast, and metastatic prostate cancer, and have demonstrated its superior performance compared to existing state-of-the-art methods.

**Abstract** Single-cell RNA-sequencing (scRNA-seq) coupled with robust computational analysis facilitates the characterization of phenotypic heterogeneity within tumors. Current scRNA-seq analysis pipelines are capable of identifying a myriad of malignant and non-malignant cell subtypes from single-cell profiling of tumors. However, given the extent of intra-tumoral heterogeneity, it is challenging to assess the risk associated with individual cell subpopulations, primarily due to the complexity of the cancer phenotype space and the lack of clinical annotations associated with tumor

scRNA-seq studies. To this end, we introduce SCellBOW, a scRNA-seq analysis framework inspired by document embedding techniques from the domain of Natural Language Processing (NLP). SCellBOW is a novel computational approach that facilitates effective identification and high-quality visualization of single-cell subpopulations. We compared SCellBOW with existing best practice methods for its ability to precisely represent phenotypically divergent cell types across multiple scRNA-seq datasets, including our in-house generated human splenocyte and matched peripheral blood mononuclear cell (PBMC) dataset. For tumor cells, SCellBOW estimates the relative risk associated with each cluster and stratifies them based on their aggressiveness. This is achieved by simulating how the presence or absence of a specific cell subpopulation influences disease prognosis. Using SCellBOW, we identified a hitherto unknown and pervasive AR−/NE$_{low}$ (androgen-receptor-negative, neuroendocrine-low) malignant subpopulation in metastatic prostate cancer with conspicuously high aggressiveness. Overall, the risk-stratification capabilities of SCellBOW hold promise for formulating tailored therapeutic interventions by identifying clinically relevant tumor subpopulations and their impact on prognosis.

## Introduction

Intra- and inter-tumoral heterogeneity are pervasive in cancer and manifest as a constellation of molecular alterations in tumor tissues. The late-stage clonal proliferation, partial selective sweeps, and spatial segregation within the tumor mass collectively orchestrate lineage plasticity and metastasis (*Dentro et al., 2021*). In collaboration with non-malignant cell types in the tumor stroma, malignant cells with distinct genetic and phenotypic properties create complex and dynamic ecosystems, rendering the tumors recalcitrant to therapies (*Lawson et al., 2018*). Thus, the phenotypic characterization of malignant cell subpopulations is critical to understanding the underlying mechanisms of resistive behavior. The widespread adoption of single-cell RNA-sequencing (scRNA-seq) has enabled the profiling of individual cells, thereby obtaining a high-resolution snapshot of their unique molecular landscapes (*Bhattacharya et al., 2021*; *Kanev et al., 2021*). A precise understanding of cell-to-cell functional variability captured by scRNA-seq profiles is crucial in this context. These molecular profiles assist in robust deconvolution of the oncogenic processes instigated by various selection pressures exerted by anticancer agents. They also facilitate understanding the cross-talks between malignant and non-malignant cell types within the tumor microenvironment (*Dagogo-Jack and Shaw, 2018*). To effectively analyze tumor scRNA-seq data, various specialized techniques have been developed. These techniques assist in proactive investigation of complex and elusive cell populations (*Pang et al., 2024*; *Poonia et al., 2023*), regulatory gene interactions (*Chapman et al., 2022*), neoplastic cell lineage trajectories (*Simeonov et al., 2021*), and expression-based inference of copy number variations (*Tickle et al., 2019*). While these computational techniques have been successful in gaining novel biological insights, their adoption in clinical settings is still elusive.

Over the past years, leading consortia such as The Cancer Genome Atlas (TCGA; *Weinstein et al., 2013*) and other large-scale independent studies have established reproducible molecular subtypes of cancers with divergent prognoses. For instance, metastatic prostate cancer has typically been categorized based on the androgen receptor (AR) activity or neuroendocrine (NE) program: the less aggressive AR+/NE− (AR prostate cancer, ARPC) and the highly aggressive AR−/NE+ (NE prostate cancer, NEPC; *Beltran et al., 2016*). More recent studies have identified additional phenotypes, such as the AR−/NE− (double-negative prostate cancer, DNPC) and AR+/NE+ (amphicrine prostate cancer, AMPC) (*Brady et al., 2021*; *Han et al., 2022*). These findings underscore the importance of considering tumor heterogeneity while dictating the differential treatment regimes to improve patient outcomes (*Chawla et al., 2022*). These studies are predominantly based on bulk omics assays, which precludes the detectability of fine-grained molecular subtypes of clinical relevance. To address this, there is an urgent need to develop novel analytical approaches that are capable of exploiting single-cell omics profiles for risk attribution to malignant cell subtypes.

A growing number of NLP-based methods have recently been gaining popularity for their ability to predict patients' survival based on their clinical records and for facilitating efficient analyses of high-dimensional scRNA-seq data (*Nunez et al., 2023*; *Kim et al., 2021*; *Zhao et al., 2021*; *Yang et al., 2022*). For instance, (*Nunez et al., 2023*) employ language models to predict the survival outcomes of breast cancer patients based on their initial oncologist consultation documents. Similarly, in the

domain of scRNA-seq analysis, scETM (*Zhao et al., 2021*) uses topic models to infer biologically relevant cellular states from single-cell gene expression data. Recently, scBERT (*Yang et al., 2022*), an adaptation of attention-based transformer architecture, has been developed for cell type annotation in single-cell. Motivated by the objective to efficiently associate survival risk directly with cellular subpopulations extracted from scRNA-seq data, we introduce SCellBOW (single-cell bag-of-words), a Doc2vec (*Le and Mikolov, 2014*) inspired transfer learning framework for single-cell representation learning, clustering, visualization, and relative risk stratification of cell types within a tumor microenvironment. SCellBOW intuitively treats cells as documents and genes as words. SCellBOW learned latent representations capture the semantic meanings of cells based on their gene expression levels. Due to this, cell type or condition-specific expression patterns get adequately captured in cell embeddings. We demonstrate the utility of SCellBOW in clustering cells based on their semantic similarities across multiple scRNA-seq datasets, spanning from normal prostate to pancreas and peripheral blood mononuclear cells (PBMC). As an extended validation, we apply SCellBOW to an in-house scRNA-seq dataset comprising human PBMC and matched splenocytes. We observed that SCellBOW outperforms existing best-practice single-cell clustering methods in its ability to precisely represent phenotypically divergent cell types.

Beyond robust identification of cellular clusters, the latent representation of single cells provided by SCellBOW captures the 'semantics' associated with cellular phenotypes. In line with *word algebra* supported by NLP models, which explores word analogies based on semantic closeness (e.g. adding a word vector associated *royal* to *man* brings it closer to *king*), we conjectured that cellular embedding could reveal biologically intuitive outcomes through algebraic operations. Thus, we aimed to replicate this feature in the single-cell phenotype space to introduce *phenotype algebra* and apply the same to attribute prognostic value to cancer cell subpopulations identified from scRNA-seq studies. With the help of *phenotype algebra,* it is possible to simulate the exclusion of the phenotype associated with a specific subpopulation within the tumor microenvironment and relate the same to the disease outcome. Selectively removing a specific subtype from the whole tumor allows us to categorize the aggressiveness of individual subtypes under negative selection pressure while preserving the impact of the residual tumor. This information can ultimately aid therapeutic decision-making and improve patient outcomes. As a proof of concept, we validate the *phenotype algebra* module of SCellBOW for risk-stratification of malignant and non-malignant cell subtypes across three independent cancer types: glioblastoma multiforme, breast cancer, and metastatic prostate cancer.

Finally, we demonstrate the utility of SCellBOW to uncover and define novel subpopulations in a cancer type based on biological features relating to disease aggressiveness and survival probability. Metastatic prostate cancer encompasses a range of malignant cell subpopulations, and characterizing novel or more fine-grained subtypes with clinical implications on patient prognosis is an active field of research (*Kfoury et al., 2021*; *Feng et al., 2022*). To explore this, we applied our *phenotype algebra* on the SCellBOW clusters as opposed to predefined subtypes. Through our analysis using SCellBOW, we identify a dedifferentiated metastatic prostate cancer subpopulation. This subpopulation is nearly ubiquitous across numerous metastatic prostate cancer samples and demonstrates greater aggressiveness compared to any known molecular subtypes. To the best of our knowledge, SCellBOW pioneered the application of a NLP-based model to attribute survival risk to malignant and non-malignant cell subpopulations from patient tumors. SCellBOW holds promise for tailored therapeutic interventions by identifying clinically relevant subpopulations and their impact on prognosis.

## Results
### Overview of SCellBOW

Correlating genomic readouts of tumors with clinical parameters has helped us associate molecular signatures with disease prognosis (*Abida et al., 2019*). Given the prominence of phenotypic heterogeneity in tumors, it is important to understand the connection between molecular signatures of cellular populations and disease aggressiveness. Existing methods map tumor samples to a handful of well-characterized molecular subtypes with known survival patterns primarily obtained from bulk RNA-seq studies. However, bulk RNA-seq measures the average level of gene expression distributed across millions of cells in a tissue sample, thereby obscuring the intra-tumoral heterogeneity (*Zhu et al., 2017*). This limitation prompted us to turn to tumor scRNA-seq. Tumor scRNA-seq studies

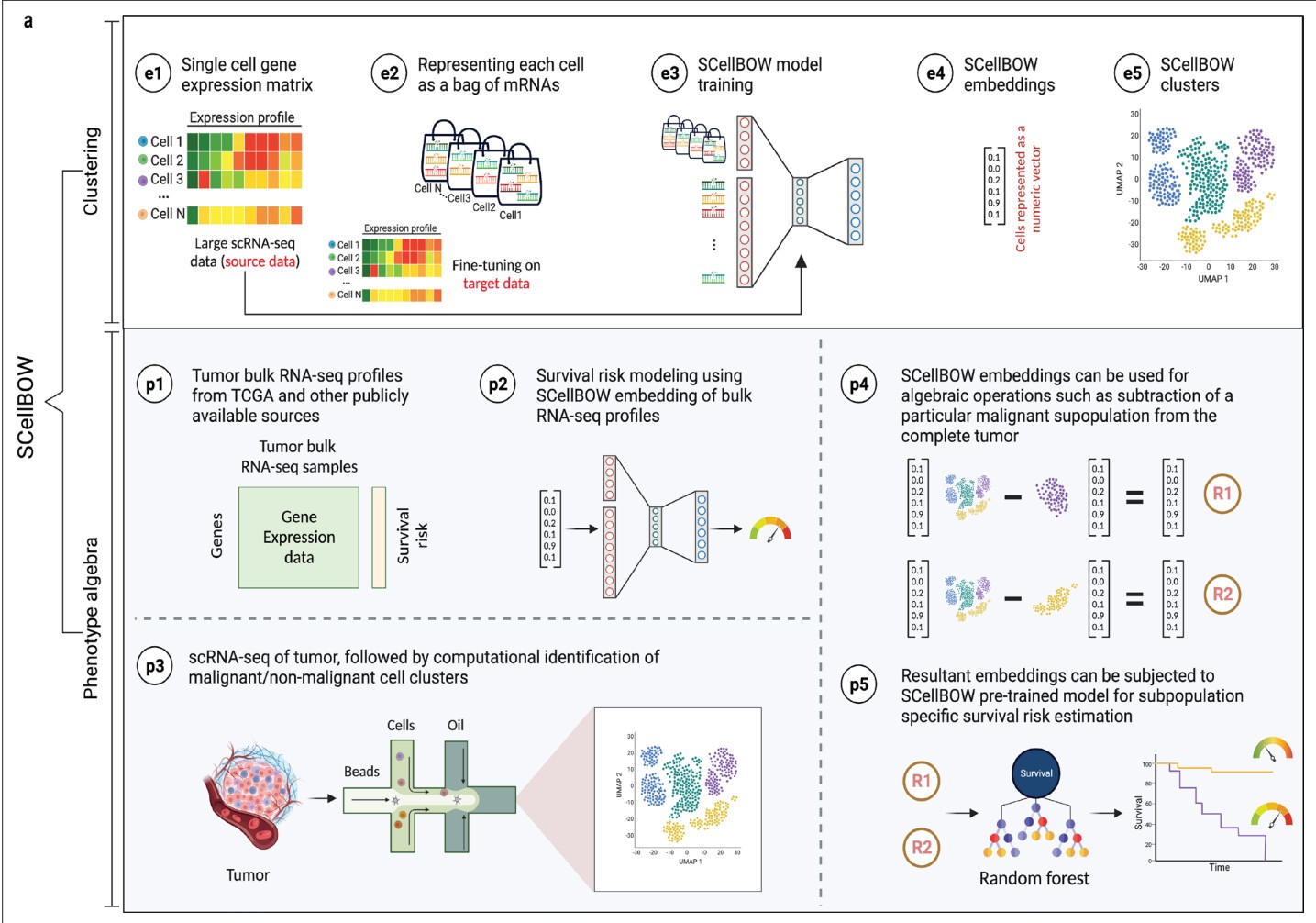

**Figure 1.** SCellBOW workflow. (**a**) Schematic overview of SCellBOW workflow for identifying cellular clusters and assessing the aggressiveness of the predicted clusters. For SCellBOW clustering, firstly, a corpus was created from the gene expression matrix, where cells were analogous to documents and genes to words. Next, the pre-trained model was retrained with the vocabulary of the target dataset. Then, clustering was performed on embeddings generated from the neural network. For SCellBOW *phenotype algebra*, vectors were created for reference (*total tumor*) and queries. Then, the query vector was subtracted from the reference vector to calculate the predicted risk score using a bootstrapped random survival forest. Finally, survival probability was evaluated, and phenotypes were stratified by the median predicted risk score. Created using BioRender.com.

have successfully revealed the extent of gene expression variance across individual malignant cells, contributing to an in-depth understanding of the mechanisms driving cancer progression (**Baslan and Hicks, 2017**). However, to date, most studies use marker genes identified from bulk RNA-seq studies to characterize malignant cell clusters identified from scRNA-seq data. This approach is inadequate since every tumor is unique when looked through the lenses of single-cell expression profiles (**Zhang et al., 2021**). Bulk markers alone cannot adequately capture the intricate heterogeneity present within tumors, as different tumors may exhibit distinct phenotypes with varying degrees of aggressiveness at the single-cell level. While existing methods effectively reveal the subpopulations, they are insufficient in associating malignant risk with specific cellular subpopulations identified from scRNA-seq data. The proposed approach, SCellBOW, can effectively capture the heterogeneity and risk associated with each phenotype, enabling the identification and assessment of malignant cell subtypes in tumor microenvironment directly from scRNA-seq gene expression profiles, thereby eliminating the need for marker genes (**Figure 1**). Below, we highlight key constructions and benefits of this new approach.

SCellBOW adapts the popular document-embedding model Doc2vec for single-cell latent representation learning, which can be used for downstream analysis. SCellBOW represents cells as documents and gene names as words. Gene expression levels are encoded by 'word frequencies', that

is variation in gene expression values is captured by introducing gene names in proportion to their intensities in cells. SCellBOW encodes each cell into low-dimensional embeddings. SCellBOW extracts the gene expression patterns of individual cells from a relatively large unlabeled source scRNA-seq dataset by pre-training a shallow source network. The neuronal weights estimated during pre-training are transferred to a relatively smaller unlabeled scRNA-seq dataset, where they are refined based on the gene expression patterns in the target dataset. SCellBOW leverages the variability in gene expression values to subsequently cluster cells according to their semantic similarity.

Building upon capability of SCellBOW to preserve the semantic meaning of individual cells, we established meaningful associations among cellular phenotypes. SCellBOW offers a remarkable feature for executing algebraic operations such as '+' and '–' on single cells in the latent space while preserving the biological meanings. This feature catalyzes the simulation of the residual phenotype of tumors, following positive and negative selection of specific cell subtype in a tumor. This could potentially be used to identify the contribution of that phenotype to patient survival. For example, the '–' operation can be used to predict the likelihood of survival by eliminating the impact of a specific aggressive cellular phenotype from the whole tumor. We empirically show the retention of such semantic relationships in the context of cellular phenotypes of cancer cells using *phenotype algebra*. SCellBOW uses algebraic operations to compare and analyze the importance of cellular phenotypes working independently or in combination toward the aggressiveness of the tumor in a way that is not possible using traditional methods. *Phenotype algebra* can be performed either on the pre-defined cancer subtypes or SCellBOW clusters. Drawing from the overall performance of SCellBOW in accurately clustering and ranking cellular phenotypes by aggressiveness, we further analyzed multi-patient scRNA-seq data of metastatic prostate cancer and characterized an unknown, de-differentiated AR−/ $NE_{low}$ subpopulation of malignant cells.

## SCellBOW effectively captures latent space of single-cell phenotypes

Malignant cells are far more heterogeneous compared to associated normal cells. Clustering is often the first step toward recognizing cellular lineages in a tumor sample. Clustering single cells based on their molecular profiles can potentially identify rare cell populations with distinct phenotypes and clinical outcomes. To evaluate the strength of the clustering ability of SCellBOW, we benchmarked our method against five existing scRNA-seq clustering methods (*Zhao et al., 2021*; *Butler et al., 2018*; *Wolf et al., 2018*; *Li et al., 2020*; *Hu et al., 2020*; *Appendix 1—table 1*). Among these packages, Seurat (*Butler et al., 2018*) and Scanpy (*Wolf et al., 2018*) are the most popular, and both employ graph-based clustering techniques. DESC (*Li et al., 2020*) is a deep neural network-based scRNA-seq clustering package, whereas ItClust (*Hu et al., 2020*) and scETM (*Zhao et al., 2021*) are transfer learning methods. All the packages are resolution-dependent except for ItClust. ItClust automatically selects the resolution with the highest silhouette score. We used a number of scRNA-seq datasets to evaluate the methods cited above (*Appendix 1—table 2*). For the objective evaluation of performance, we used an adjusted Rand index (ARI) and normalized mutual information (NMI) to compare clusters with known cell annotations (*Appendix 1—table 3*). For all methods, except for ItClust, we computed overall ARI and NMI for different resolution values ranging between 0.2 and 2.0. We computed the cell type silhouette index (SI) based on known annotations to measure the signal-to-noise ratio of low-dimensional single-cell embeddings.

We constructed three use cases leveraging publicly available scRNA-seq datasets. Each instance constitutes a pair of single-cell expression datasets of which the source data is used for self-supervised model training and the target data for model fine-tuning and analysis of the clustering outcomes. In all cases, the target data has associated cell type annotations derived from fluorescence-activated cell sorting (FACS) enriched pure cell subpopulations. The first use case consists of non-cancerous cells from prostate cancer patients (*Karthaus et al., 2020*; 120,300 cells) as the source data and cells from healthy prostate tissues (*Henry et al., 2018*; 28,606 cells) as the target data. This use case was designed to assess the resiliency of SCellBOW to the presence of disease covariates in a large scRNA-seq dataset. The second use case comprises a large PBMC dataset (*Zheng et al., 2017*; 68,579 cells) as the source data, whereas the target data was sourced from a relatively small FACS-annotated PBMC dataset (2700 cells) from the same study. The third use case, as source data, comprises pancreatic cells from three independent studies processed with different single-cell profiling technologies (inDrop *Baron et al., 2016*, CEL-Seq2 *Muraro et al., 2016*, SMARTer *Wang et al., 2016*). The target data

used in this case was from an independent study processed with a different technology, Smart-Seq2 (*Segerstolpe et al., 2016*; *Figure 2*, *Appendix 2—figure 1*). For the normal prostate, PBMC, and pancreas datasets, SCellBOW produced the highest ARI scores across most notches of the resolution spectrum (0.2–2.0; *Figure 2g–i*). We observed a similar trend in the case of NMI (*Figure 2j*). SCellBOW exhibited the highest NMI compared to other methods for the normal prostate, PBMC, and pancreas datasets. For further deterministic evaluation of the different methods across the datasets, we set 1.0 as the default resolution for calculating cell type SI (*Figure 2k*). In the PBMC and pancreas datasets, SCellBOW yielded the highest cell type SI for both datasets. SCellBOW and Seurat were comparable in performance for the pancreas dataset, outperforming other methods. We observed poor performance by DESC and ItClust across all the datasets in terms of cell type SI. In terms of cell type SI, Seurat and scETM, showed improved results in the normal prostate dataset. However, SCellBOW outperformed both Seurat and scETM in terms of overall ARI and NMI for the same dataset, indicating higher clustering accuracy. To further evaluate the cluster quality in the normal prostate dataset, we compared the known cell types to the predicted clusters. Known cell types such as basal epithelial, luminal epithelial, and smooth muscle cells were grouped into homogeneous clusters by SCellBOW (*Appendix 2—figure 2*). We observed that the majority of fibroblasts and endothelial cells were mapped by SCellBOW to single clusters, unlike Seurat and scETM. SCellBOW retained hillock cells in close proximity to both basal epithelial and club cells, in contrast to Seurat, which only includes basal epithelial cells. We compared SCellBOW with two additional methods– scBERT (*Yang et al., 2022*) and scPhere (*Ding and Regev, 2021*). Among these, scBERT is a transfer learning-based method built upon BERT (*Devlin et al., 2018*). scPhere is a deep generative model designed for embedding single cells into low-dimensional hyperspherical or hyperbolic spaces. While scBERT offers competitive performance, scPhere consistently falls behind the other methods (*Appendix 2—figure 3*, Appendix 2—note 1).

Our analyses of the public datasets confirm the robustness of SCellBOW compared to the prominent single-cell analysis methods, including the prominent transfer learning methods. To this end, we applied SCellBOW to investigate a more challenging task of analyzing an in-house scRNA-seq data comprising splenocytes and matched PBMCs from two healthy and two brain-dead donors (*Figure 3a*). Given multiple covariates, such as the origin of the cells and the physiological states of the donors, analysis of this scRNA-seq data presents a challenging use case (*Figure 3b and c*). We used the established high-throughput scRNA-seq platform CITE-seq (*Stoeckius et al., 2017*) to pool eight samples into a single experiment. After post-sequencing quality control, we were left with 4819 cells. We annotated the cells using Azimuth (*Hao et al., 2021*), with occasional manual interventions (*Figure 3d*, *Appendix 2—figure 4d–f*, Appendix 2—note 2). We quantitatively evaluated SCellBOW and the rest of the benchmarking methods by measuring ARI, NMI, and cell type SI (*Figure 3e*, *Appendix 2—figure 3d–f*). While most methods did reasonably well, SCellBOW offered an edge. We observed the best results in SCellBOW in terms of ARI, NMI, and cell type SI compared to other methods (*Appendix 1—table 3*). SCellBOW yielded clusters largely coherent with the independently done cell type annotation using Azimuth (*Figure 3*, *Appendix 2—figure 5*). Further, most clusters harbor cells from all the donors, indicating that the sample pooling strategy was effective in reducing batch effects. B cells, T cells, and NK cells map to SCellBOW clusters where the respective cell types are the majority. Most CD4 +T cells map to CL0 and CL9 (here, CL is used as an abbreviation for cluster; *Figure 3f*). CL0 is shared between CD8 +T cells, CD4 +T cells, and Treg cells, which originate from the same lineage. CL4 majorly consists of NK cells with a small fraction of CD8 +T cells, which is not unduly deviant from the PBMC lineage tree. As a control, we performed similar analyses of the Scanpy clusters (*Figure 3g*). While Scanpy performed reasonably well, misalignments could be spotted. For example, CD4 +and CD8+T cells were split across many clusters with mixed cell type mappings. SCellBOW maps CD14 monocytes to a single cluster, whereas Scanpy distributes CD14 monocytes across two clusters (CL1 and CL8), wherein CL8 is equally shared with conventional Dendritic Cells (cDC). 'Eryth' annotated cells are indignantly mapped to different clusters by both SCellBOW and Scanpy. This could be due to the Azimuth's reliance on high mitochondrial gene expression levels for annotating erythroid cells (Appendix 2—note 3). However, SCellBOW CL6 is dominated by Erythroid cells with marginal interference from other cell types. In the case of Scanpy, no such cluster could be detected. Discerning phenotypic heterogeneity from the expression profiles of seemingly similar cells is a challenging task. The performance of SCellBOW, with near-ground-truth cell type annotation, in

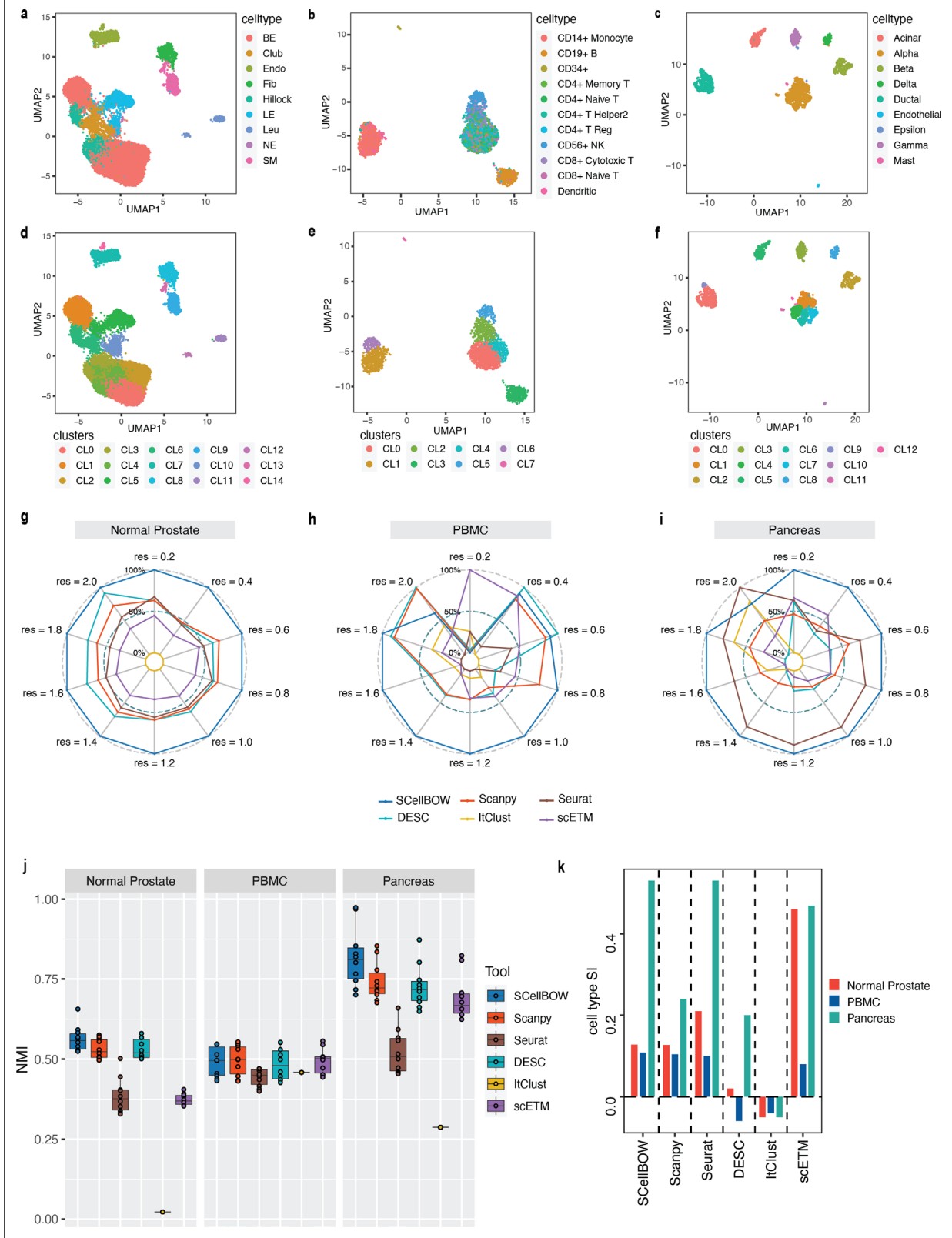

**Figure 2.** Evaluation of single-cell representations using SCellBOW. (**a–c**) UMAP plots for the normal prostate (**a**), PBMC (**b**), and pancreas (**c**) datasets. The coordinates are colored by cell types. (**d-f**) UMAP plots for normal prostate (**d**), PBMC (**e**), and pancreas (**f**) datasets, where the coordinates are colored by SCellBOW clusters. CL is used as an abbreviation for cluster. (**g–i**) Radial plot for the percentage of contribution of different methods towards ARI for various resolutions ranging from 0.2 to 2.0. ItClust is a resolution-independent method; thus, the ARI is kept constant across all the resolutions. (**j**)

*Figure 2 continued on next page*

*Figure 2 continued*

Box plot for the NMI of different methods across different resolutions ranging from 0.2 to 2.0 in steps of 0.2. (**k**) Bar plot for the cell type silhouette index (SI) for different methods. The default resolution was set to 1.0.

such a scenario confirmed its ability to adequately decipher the underlying cellular heterogeneity and provide robust cell type clustering.

## SCellBOW enables survival risk attribution of tumor subpopulations

Every cancer features unique genotypic as well as phenotypic diversity, impeding the personalized management of the disease. The widespread adoption of genomics in cancer care has allowed correlating molecular portraits of tumors with patient survival in all major cancers. A few studies suggested that there is an association between the tumorigenicity of stromal cells in tumor microenvironments and patient survival (*Dwivedi et al., 2022*). This has sparked debate about the utility of single-cell gene expression in profiling single tumor cells. The *phenotype algebra* module of SCellBOW estimates the relative aggressiveness of different malignant and non-malignant cell subtypes using survival information as a surrogate. This presents an opportunity to gauge the aggressiveness of each cancer cell subtype in a tumor. For example, subtracting an aggressive phenotype from the *total tumor* (average of SCellBOW embeddings across all cells in a tumor) would better the odds of survival relative to dropping a subtype under negative selection pressure. By associating the embedding vectors, representing *total tumor – a specific cell cluster,* with tumor aggressiveness, *phenotype algebra* immediately opens a way to infer the level of aggressiveness of a particular cluster of cancer cells obtained through single-cell clustering.

As proof of concept, we first validated our approach on glioblastoma multiforme (GBM), which has been studied widely employing single-cell technologies. GBM has three well-characterized malignant subtypes: proneural (PRO), classical (CLA), and mesenchymal (MES; *Tang et al., 2021*; *Behnan et al., 2019*). We obtained known markers of PRO, CLA, and MES to annotate 4508 malignant GBM cells obtained from a single patient reported by Couturier and colleagues (*Couturier et al., 2020*) and used it as our target data (*Figure 4a*; for details on cell type annotation, refer to Appendix 1, Supplementary methods 2.1). As the source data, we used the GBM scRNA-seq data from *Neftel et al., 2019*. The survival data consisted of 613 bulk GBM samples with paired survival information from the TCGA consortium. We constructed the following query vectors for the survival prediction task: *total tumor*, i.e., average of embeddings of all malignant cells, *total tumor – (MES +CLA)*, *total tumor – MES/CLA/PRO* (individually). We conjectured that for the most aggressive malignant cell subtype, *total tumor – subtype specific pseudo-bulk*, would yield the biggest drop in survival risk relative to the *total tumor*. Survival risk predictions associated with *total tumor – MES/CLA/PRO* thus obtained reaffirmed the clinically known aggressiveness order, that is CLA > MES > PRO, where CLA succeeds the rest of the subtypes in aggressiveness (*Lin et al., 2014*; *Figure 4c and d*). More complex queries can be formulated, such as *total tumor – (MES +CLA)*, which indicates that the tumor does not comprise the two most aggressive phenotypes, CLA and MES, and instead consists only of PRO cells. Hypothetically, this represents the most favorable scenario, as testified by the *phenotype algebra* (*Behnan et al., 2019*; *Patel et al., 2014*).

We performed a similar benchmarking on well-established PAM50-based (*Parker et al., 2009*) breast cancer (BRCA) subtypes: luminal A (LUMA), luminal B (LUMB), HER2-enriched (HER2), basal-like (BASAL), and normal-like (NORMAL; *Figure 4b*). We used 24,271 cancer cells from *Wu et al., 2020* as the source data, 545 single-cell samples from *Zhou et al., 2021* as the target data. We used 1,079 bulk BRCA samples with paired survival information from TCGA as the survival data. SCellBOW-predicted survival risks for the different subtypes were generally in agreement with the clinical grading of the PAM50 subtypes (*Mathews et al., 2019*; *Weigelt et al., 2010*). Exclusion of LUMA from *total tumor* yielded the highest risk score, indicating that LUMA has the best prognosis, followed by HER2 and LUMB, whereas BASAL and NORMAL were assigned worse prognosis (*Hennigs et al., 2016*; *Ahn et al., 2015*; *Yersal and Barutca, 2014*; *Figure 4e and f*). We observed an interesting misalignment from the general perception about the relative aggressiveness of the NORMAL subtype- removal of this subtype from *total tumor* indicated the highest improvement in prognosis. The NORMAL subtype is a poorly characterized and rather heterogeneous category. Recent evidence suggests that these

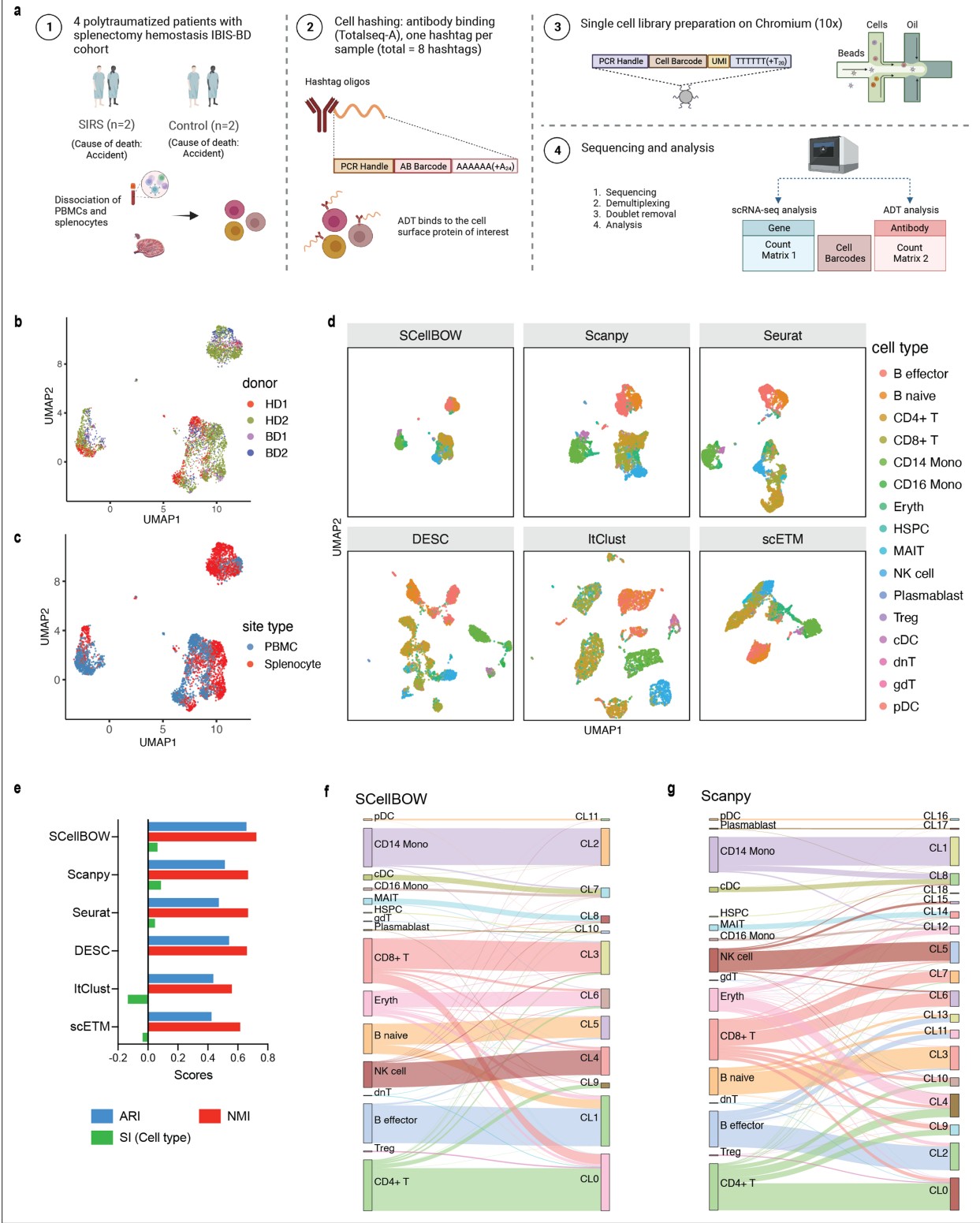

**Figure 3.** Evaluation of in-house splenocytes and matched PBMC dataset. (**a**) An experiment schematic diagram highlighting the sites of the organs for tissue collection and sample processing. In this matched PBMC-splenocyte CITE-seq experiment, PBMCs and splenocytes were collected, followed by high-throughput sequencing and downstream analyses. Created using BioRender.com. (**b, c**) UMAP plots for SCellBOW embedding colored by donors (**b**) and cell types (**c**). (**d**) The UMAP plots for the embedding of SCellBOW compared to different benchmarking methods. The coordinates of all the plots are colored by cell type annotation results using Azimuth. (**e**) Bar plot for ARI, NMI, cell type SI at resolution 1.0. (**f, g**) Alluvial plots for Azimuth cell types mapped to SCellBOW clusters (**f**) and Scanpy clusters (**g**). The resolution of SCellBOW was set to 1.0. CL is used as an abbreviation for cluster.

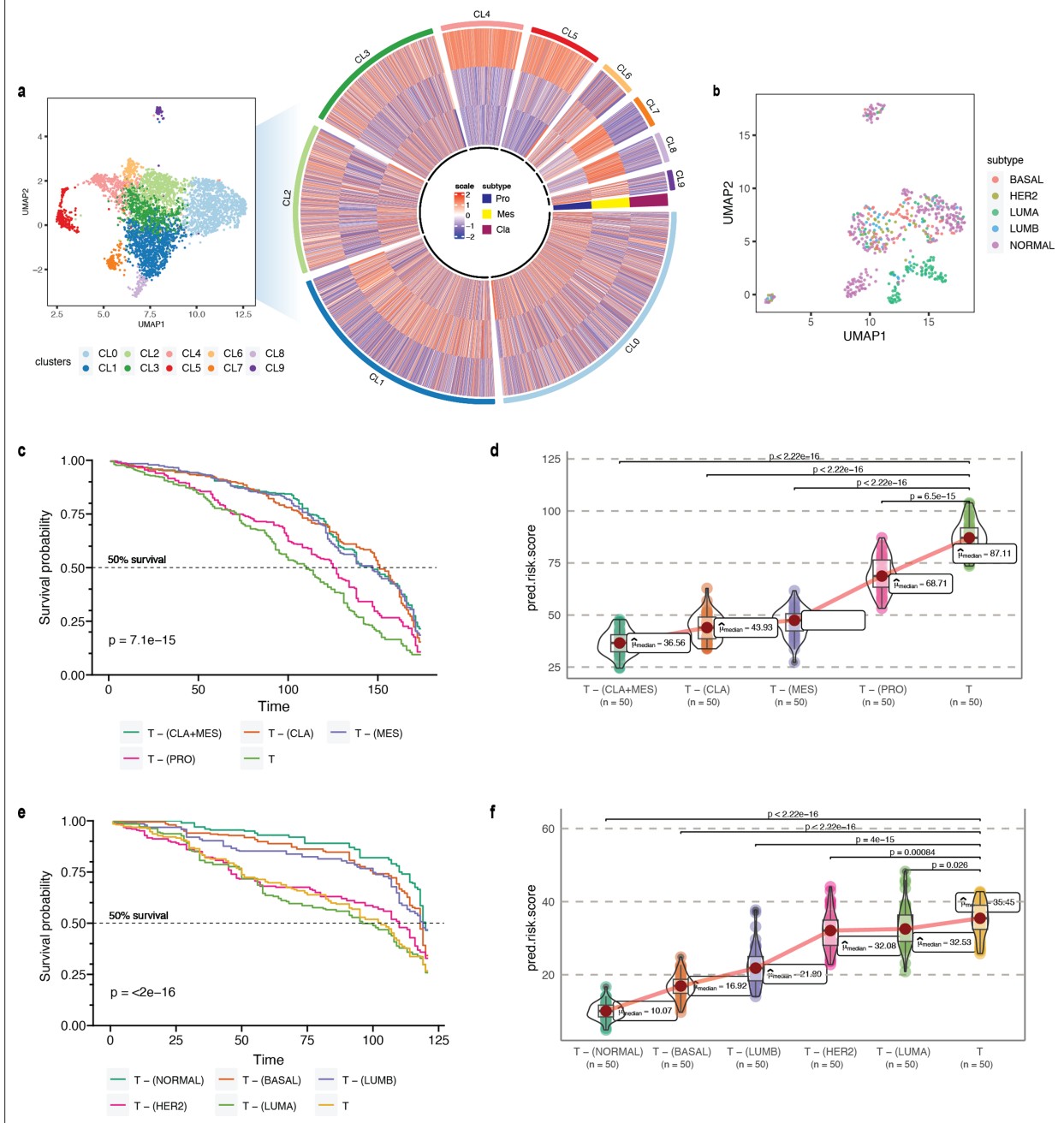

**Figure 4.** *Phenotype algebra* on GBM and BRCA known molecular subtypes. (**a**) Heatmap for GSVA score for three molecular subtypes of GBM: CLA, MES, and PRO, grouped by SCellBOW clusters at resolution 1.0. (**b**) UMAP plot for the embedding of BRCA target dataset colored by PAM50 molecular subtype. (**c**) Survival plot for GBM molecular subtypes based on *phenotype algebra*. (**d**) Violin plot for predicted risk scores for GBM molecular subtypes, with n = 50 bootstrapped models per subtype. (**e**) Survival plot for BRCA molecular subtypes based on *phenotype algebra*. The *total tumor* is denoted by *T*. (**f**) Violin plot for predicted risk scores for BRCA molecular subtypes with n = 50 bootstrapped models per subtype.

tumors potentially represent an aggressive molecular subtype and are often associated with highly aggressive claudin-low tumors (*Mathews et al., 2019*; *Liu et al., 2014*).

To evaluate the dependence of *phenotype algebra* on SCellBOW embeddings, we first applied *phenotype algebra* directly to the raw gene expression data instead of the SCellBOW-derived embeddings. The results demonstrated that raw gene expression data alone was insufficient for accurate risk stratification (*Appendix 2—figure 6a and b*; further description in Appendix 2—note 4). To further validate the efficiency of SCellBOW in learning meaningful single-cell latent representation, we

compared its fixed-length embeddings for *phenotype algebra* to those generated by other methods-scETM, scBERT, and scPhere (*Appendix 2—figure 7a–f*). Our results showed that SCellBOW learned latent representation of single cells accurately captures the 'semantics' associated with cellular phenotypes. Whereas scETM, scBERT, and scPhere fail to stratify cancer clones in terms of their aggressiveness and contribution to disease prognosis. These benchmarking studies on the well-characterized cancer subtypes of GBM and BRCA affirm capability of SCellBOW to preserve the desirable characteristics in the resulting phenotypes obtained as outputs of algebraic expressions involving other independent phenotypes and operators such as +/–.

## SCellBOW enables accurate phenotype stratification across tumor microenvironment

In prostate cancer, the processes of transdifferentiation and dedifferentiation are vital in metastasis and treatment resistance (*Gupta et al., 2019*; *Figure 5a*). Prostate cancer originates from secretory prostate epithelial cells, where AR, a transcription factor regulated by androgen, plays a key role in driving the differentiation (*Formaggio et al., 2021*; *Stelloo et al., 2015*). Androgen-targeted therapies (ATTs) constitute the primary treatment options for metastatic prostate cancer, and they are most effective in well-differentiated prostate cancer cells with high AR activity. After prolonged treatment with ATTs, the cancers eventually progress towards metastatic castration-resistant prostate cancer (mCRPC), which is highly recalcitrant to therapy (*Einstein et al., 2021*). In response to more potent ATTs, the prostate cancer cells adapt to escape reliance on AR with low AR activity. The loss of differentiation pressure results in altered states of lineage plasticity in prostate tumors (*Antonarakis, 2019*; *Beltran et al., 2019*). The most well-defined form of treatment-induced plasticity is neuroendocrine transformation. NEPC is highly aggressive, that often manifests with visceral metastases and currently lacks effective therapeutic options (*Antonarakis, 2019*; *Yamada and Beltran, 2021*). Recent studies have pointed toward the existence of additional prostate cancer phenotypes that emerge through lineage plasticity and metastasis of malignant cells (*Han et al., 2022*). This includes malignant phenotypes such as low AR signaling and DNPC, which lack AR activity and NE features. These additional phenotypes, resulting from the mechanisms of resistance to AR inhibition, can likewise be characterized by distinct gene expression patterns. Presumably, these phenotypes represent an intermediate or transitory state of the progression trajectory from high AR activity to neuroendocrine transdifferentiation (*Labrecque et al., 2019*).

Here, we performed a pooled analysis of scRNA-seq target data consisting of 836 malignant cells derived from tumors collected from 11 tumors (mCRPC) (*He et al., 2021*). We initially classified cells into three categories - AR activity high (AR+/NE–, ARAH), AR activity low (AR$_{low}$/NE–, ARAL), and neuroendocrine (AR–/NE+, NE). The classification was based on the known molecular signatures associated with ARPC and NEPC genes (*Merkens et al., 2022*; *Figure 5b*; for details on cell type annotation, refer to Appendix 1, Supplementary methods 2.2). We used the pre-trained model built on the (*Karthaus et al., 2020*) dataset for transfer learning. We used 81 advanced metastatic prostate cancer patient samples with paired survival information from *Abida et al., 2019* as the survival data. We subsequently assessed the relative aggressiveness of these high-level categories using *phenotype algebra* (*Figure 5e and f*). We observed that subtracting the latent signature associated with the NE subtype from the tumor led to the largest drop in the predicted risk score, aligning with the anticipated order of survival (*Catapano et al., 2022*). In contrast, removing the ARAH subtype from the *total tumor* had a minimal impact on the predicted risk score. Furthermore, subtracting the ARAL subtype resulted in a predicted risk score between ARAH and NE, which supports the hypothesis that ARAL represents an intermediate state between ARAH and NE (*Merkens et al., 2022*). Upon further examination of the tumor prognosis by removing the subtypes in a combined state (ARAH +ARAL + NE), the *total tumor* had the highest positive improvement. The survival probability graph also followed the same order. Furthermore, when compared to other methods—scETM, scBERT, and scPhere—as well as raw gene expression data, these tools struggled to accurately stratify the subtypes based on their aggressiveness (*Appendix 2—figure 6c*, *Appendix 2—figure 7g–i*, Appendix 2—note 4).

Till now, *phenotype algebra* has been used primarily for the stratification of cancer subpopulations. The tumor microenvironment includes malignant cells as well as various non-malignant cell types. This diverse cellular composition significantly influences the response to drug treatments. While malignant cells within a tumor typically show a great extent of heterogeneity, other cell types, such as immune

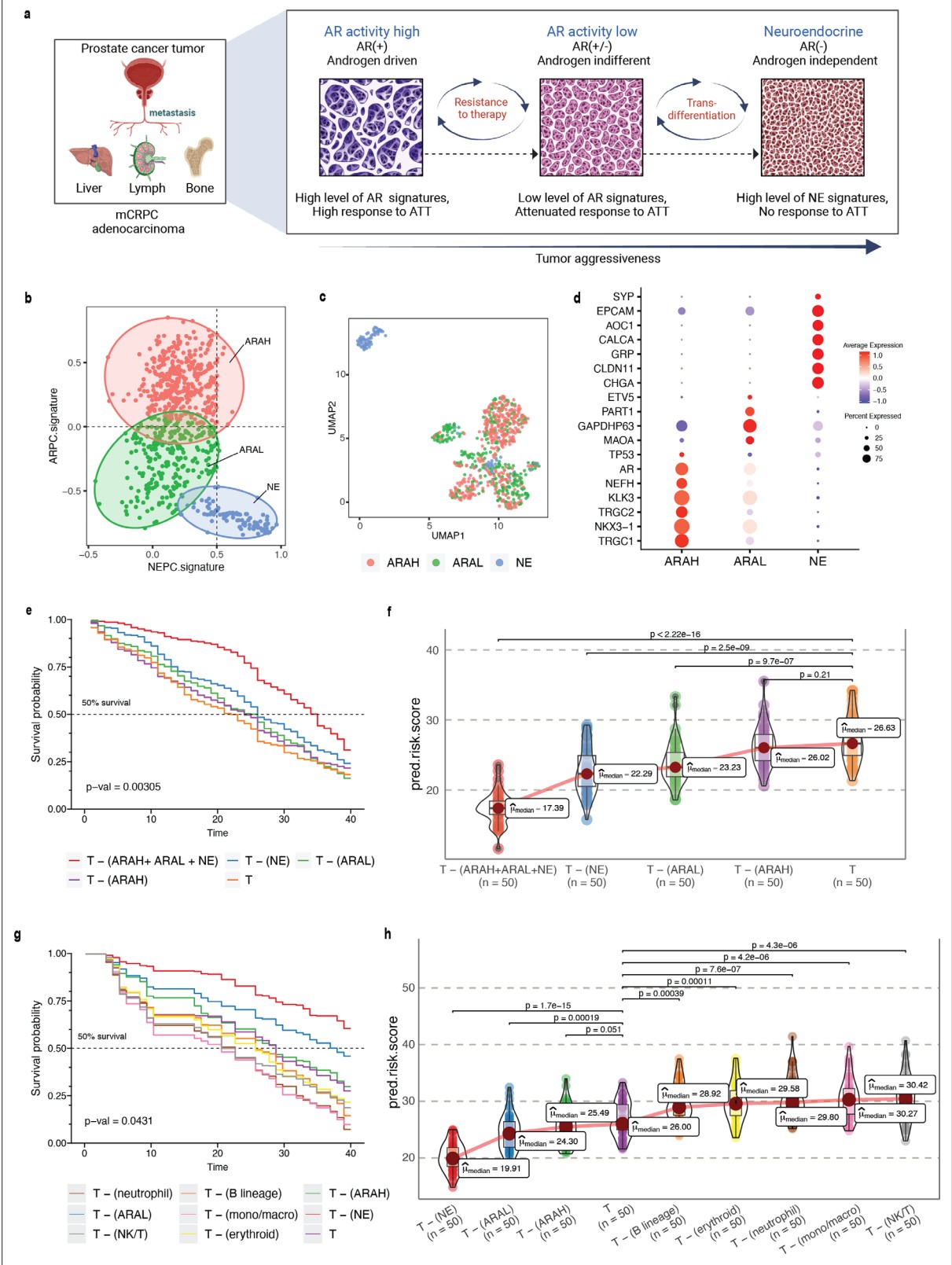

**Figure 5.** *Phenotype algebra* on mCRPC known molecular subtypes based on AR- and NE-activity. (**a**) Schematic of the transdifferentiation states underlying lineage plasticity that occurs during mCRPC progression from an ARPC to NEPC. Created using BioRender.com. (**b**) Scatter plot of GSVA scores of ARPC and NEPC gene sets, K-means clustering was used to allocate cells into the three high-level ARAH, ARAL, and NEPC categories. (**c**) UMAP plot for projection of SCellBOW embedding colored by ARAH, ARAL, and NEPC. (**d**) Heatmap showing the top differentially expressed genes (y-

*Figure 5 continued on next page*

*Figure 5 continued*

axis) between each high-level category (x-axis) and all other cells, tested with a Wilcoxon rank-sum test. (**e**) Survival plot for mCRPC cancer phenotypes based on *phenotype algebra*. The t*otal tumor* is denoted by *T*. (**f**) Violin plot for predicted risk scores for mCRPC phenotypes - ARAH, ARAL, and NEPC, with n = 50 bootstrapped models per subtype. (**g**) Survival plot for mCRPC tumor microenvironment phenotypes based on *phenotype algebra*. The t*otal tumor* is denoted by *T*. (**h**) Violin plot of predicted risk scores for mCRPC tumor microenvironment phenotypes, comparing tumor and normal cells, with n = 50 bootstrapped models per group.

cells, fibroblasts, and others, go through functional changes and variations in the amount of tumor infiltration. Therefore, it is crucial to distinguish and analyze the expression signals originating from the tumor cells in order to get a clearer picture of the gene expression changes in the cancer cells. This approach allows for a more accurate characterization of cancer subtypes based on the intrinsic properties of the malignant cells themselves. However, it is not always straightforward to unambiguously distinguish malignant from non-malignant cells in the complex environment of a tumor. To evaluate the capability of *phenotype algebra* in distinguishing between malignant and non-malignant cells, we applied it to single-cell transcriptome profiles of the mCRPC tumor microenvironment. This dataset, obtained from *He et al., 2021*, consists of 2170 cells, including malignant cells as well as various immune and stromal cell types (*Appendix 2—figure 8h*). We used the authors' annotations for cell-type classification. We observed that the predicted risk score decreases when aggressive cancer subtypes are removed from the whole tumor population (*Figure 5g and h*). Conversely, the risk score increases when immune cells are removed, suggesting that immune presence influences overall tumor aggressiveness. These results highlight the ability of *phenotype algebra* to capture and quantify risk signals from gene expression data, effectively distinguishing between malignant and non-malignant subpopulations in the tumor microenvironment.

## SCellBOW identifies malignant subpopulations with distinct risk profiles

Grouping prostate cancer cells into three high-level categories is an oversimplified view of the actual heterogeneity of advanced prostate cancer biology. Herein, SCellBOW clusters could be utilized to discover novel subpopulations based on the single-cell expression profiles that result from therapy-induced lineage plasticity. Subsequently, *phenotype algebra* can assign a relative rank to these clusters under a negative selection pressure based on their aggressiveness. We utilized this concept to cluster the malignant cells from *He et al., 2021* using SCellBOW, resulting in eight clusters, and then we predicted the relative risk for each cluster (*Figure 6a and b*). This approach enables a novel and more refined understanding of lineage plasticity states and characteristics that determine aggressiveness during prostate cancer progression. This goes beyond the conventional categorization into ARAH, ARAL, and NE.

To further elucidate these altered cellular programs, we performed a gene set variation analysis (GSVA; *Hänzelmann et al., 2013*) based on the AR- and NE- activity (Appendix 1, Supplementary methods 3, *Supplementary file 1*). Our result showed that CL4 is characterized by the highest expression of NE-associated genes and the absence of AR-regulated genes, indicating the conventional NEPC subtype. Despite CL4 having the strongest NEPC signatures, eliminating CL2 from the tumor conferred an even higher aggressiveness level. Overall, we observed that, unlike other clusters, CL2 is composed of cells from the majority of drug-treated patients and multiple metastatic sites (*Figure 6c*). This highlights that the clustering is not confounded by the individuals or the tissue origin, as often observed during integrative analysis of tumor scRNA-seq data. CL2 instead features a unique gene expression profile common to these cells. Moreover, CL2 has a mixed signature entailing ARAH, ARAL, and NE, indicating the emergence of a more transdifferentiated subtype as a consequence of therapy-induced lineage plasticity (*Figure 6d*). Even though CL2 shows NE signature, it is distinguished by the gene expression signature induced by the inactivation of the androgen signaling pathway due to ATTs. As a consequence, cells manifesting this novel signature are grouped into a single cluster (*Figure 6e*). Among all clusters, CL1 and CL3 resemble the traditional ARAH subtype. According to our *phenotype algebra* model, excluding CL6 and CL7 from the *total tumor* yielded the highest risk score. *Brady et al., 2021* have broadly partitioned metastatic prostate cancer into six phenotypic categories using digital spatial profiling (DSP) transcript and protein abundance data in spatially defined metastasis regions. To categorize the SCellBOW clusters into these broad phenotypes, we performed Pearson's

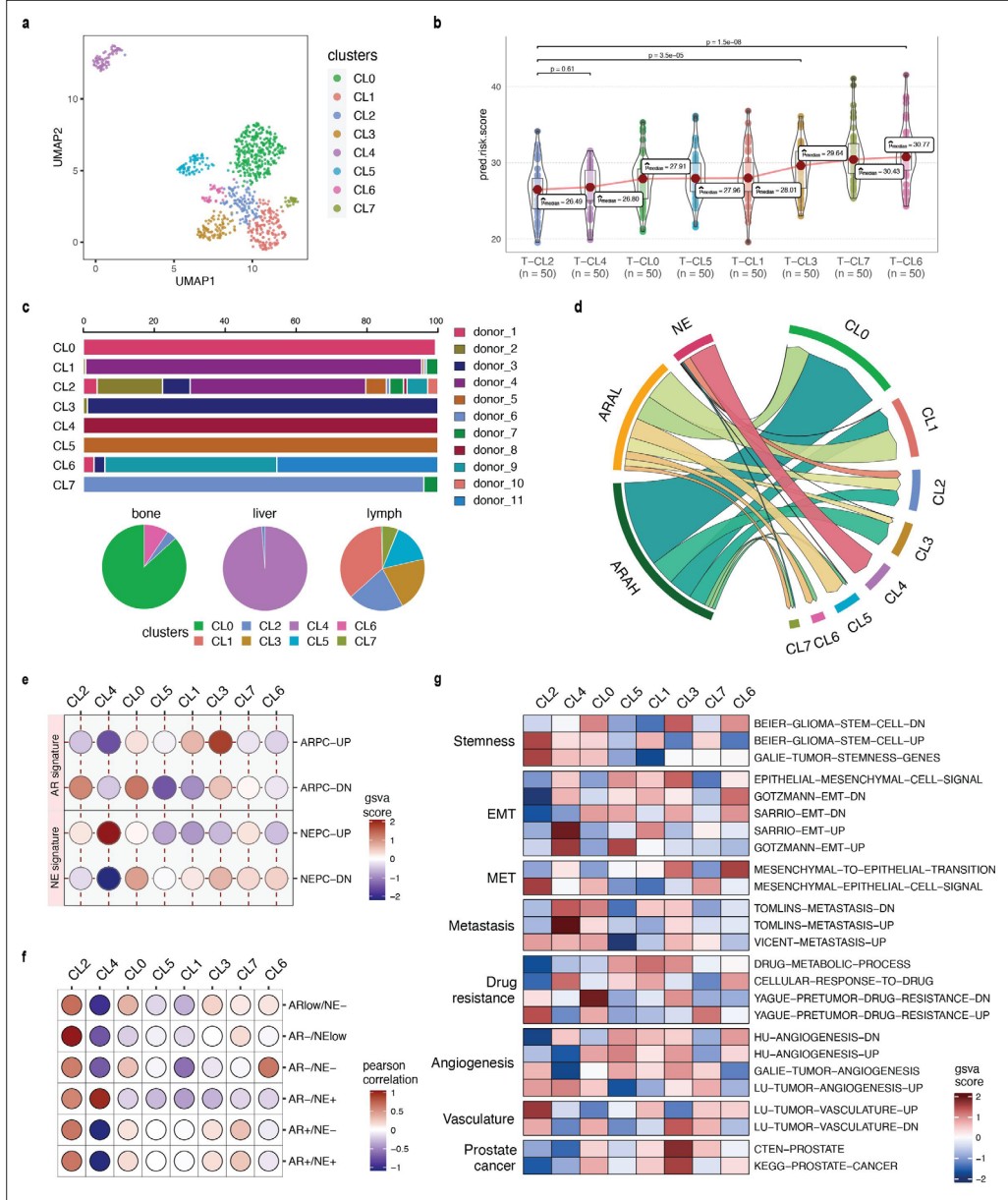

**Figure 6.** Phenotype algebra on *He et al., 2021* mCRPC data based on SCellBOW clusters. (**a**) UMAP plot for projection of embeddings with coloring based on the SCellBOW clusters at resolution 0.8. CL is used as an abbreviation for cluster. (**b**) Violin plot of *phenotype algebra*-based cluster-wise risk scores for SCellBOW clusters based on *phenotype algebra*-based predictions. (**c**) Patient and organ site distribution across the SCellBOW clusters. (**d**) Illustration of the distribution of cells from the three high-level groups- ARAH, ARAL, and NEPC across the SCellBOW clusters. (**e**) Bubble plot of row-scaled GSVA scores for custom curated gene sets containing activated and repressed AR- and NE- signatures. (**f**) Correlation plot of six phenotypic categories based on DSP gene expression correlated with the SCellBOW clusters based on scRNA-seq gene expression. The six phenotypic categories are defined by *Brady et al., 2021* based on the activity of AR and NE programs. (**g**) Top gene sets correlated with SCellBOW clusters. Signatures were collected from the C2 "curated", C5 "Gene Ontology", and H "hallmark" gene sets from mSigDB (*Liberzon et al., 2015*). Ranking by row scaled GSVA scores of one cluster against all others.

correlation test between averaged DSP expression measurements of the six phenotypes and averaged scRNA-seq expression of the SCellBOW clusters (*Figure 6f*). The results revealed that CL2 has the highest correlation with the phenotype defined by lack of expression of AR signature genes and low or heterogeneous expression of NE-associated genes (AR–/NE<sub>low</sub>). Similarly, as expected, CL4

showed the highest correlation with the NEPC phenotype defined by positive expression of NE-associated genes without AR activity (AR–/NE+). Meanwhile, CL6 exhibits a closer resemblance to a DNPC phenotype (AR–/NE–).

To gain a deeper understanding of these modified cellular processes in CL2 compared to other clusters, we conducted cluster-wise functional GSVA based on the hallmarks of cancer (*Figure 6g*). We observed that CL2 exhibited the least prostate cancer signature, indicating that this cluster has deviated from prototypical prostate cancer behavior. It has rather dedifferentiated into a more aggressive phenotype as a consequence of therapy-induced lineage plasticity (*Wang et al., 2014*; *Merkens et al., 2022*). CL2 exhibited the highest enrichment of genes related to cancer stemness compared to other clusters. CL2 showed pronounced repression of epithelial genes that are downregulated during the epithelial-to-mesenchymal transition (EMT). Furthermore, there is a lack of expression of genes that are upregulated during the reversion of mesenchymal-to-epithelial phenotype (MET). Thus, the cells in CL2 are undergoing a process of dedifferentiation from being epithelial cells and activating the mesenchymal gene networks. Existing studies have reported that adaptive resistance is positively correlated with the acquisition of mesenchymal traits in cancer (*Hangauer et al., 2017*; *Catapano et al., 2022*). CL2 was enriched with signatures associated with metastasis and drug resistance. Specifically, CL2 showed downregulation of the genes involved in drug metabolism and cellular response while upregulation of the genes associated with drug resistance. In cancer cells, the acquisition of stemness-like as well as cell dedifferentiation (mesenchymal) traits can facilitate the formation of metastases and lead to the development of drug resistance (*Castellón et al., 2022*). Further analysis of the metastatic potential of CL2 indicated that the cluster is enriched with genes associated with vasculature and angiogenesis. Tumors induce angiogenesis in the veins and capillaries of the host tissue to become vascularized, which is crucial for their growth and metastasis (*Lugano et al., 2020*). Thus, based on our findings, we anticipate that CL2 corresponds to a highly aggressive and dedifferentiated subpopulation of mCRPC within the lineage plasticity continuum, correlating to poor patient survival and positive metastatic status. Our cluster-wise *phenotype algebra* results imply that the traits of the androgen and neuroendocrine signaling axes are not the exclusive defining features of the predicted risk ranking and that other, yet under-explored biological programs play additional important roles.

## Discussion

In this work SCellBOW, a scRNA-seq analysis framework inspired by NLP-based transfer learning approach can be utilized to decipher molecular heterogeneity from scRNA-seq profiles and infer survival risks associated with the individual cell subpopulations in tumor microenvironment. In this novel approach, cells are treated as a bag-of-molecules, similar to representing a document as a 'bag' of words (BOW), representing molecules in proportion to their abundance within each cell. SCellBOW uses a document embedding architecture to preserve the cellular similarities observed in the gene expression space. SCellBOW-learned neuronal weights are transferable. The model can use the knowledge acquired from a source dataset to warm start the learning process on a target dataset. Although Doc2vec is not commonly used for transfer learning, our experiments, which involve repurposing a pre-trained, modified Doc2vec model for scRNA-seq analysis, establish promising use cases for transfer learning (*Appendix 2—figure 9*, Appendix 2—note 5). SCellBOW accomplishes two major tasks: a. single-cell representation learning under a transfer learning framework facilitating high-quality cell clustering and visualization of scRNA-seq profiles. b. *phenotype algebra*, enabling the attribution of survival risk to tumor cell clusters obtained from tumor scRNA-seq data. For both tasks, we compared SCellBOW with several state-of-the-art single-cell analysis methods, including ones based on complex language modeling architectures (e.g. scETM uses a topic model, scBERT uses a transformer-inspired model). SCellBOW exhibited consistency in characterizing cellular heterogeneity through clustering and survival risk (aggressiveness) stratification of tumor cell subtypes.

With the SCellBOW, we observe considerable improvement in the quality of clustering and *phenotype algebra* results obtained from scRNA-seq target datasets. The existing single-cell transfer learning methods, such as scETM, ItClust, and scBERT, are pre-trained on large amounts of pre-annotated single-cell data to achieve good performance. However, using source data annotations as a reference limits the identification of new cell types in the target data. Moreover, the degree of imbalance in the cell type distribution substantially influences the performance of these methods (*Khan et al., 2023*). It

is important to realize that exploratory studies involving single-cell expression profiling require unsupervised data analysis. SCellBOW addresses these challenges by allowing self-supervised pre-training on gene expression datasets, learning the general syntax and semantic patterns from the unlabeled scRNA-seq dataset. Additionally, it is less susceptible to overfitting (*Wu et al., 2021*), especially with small training datasets, owing to the shallow neural network used in SCellBOW, which enables faster convergence compared to other studied transfer learning methods (*Appendix 1—table 5*). Notably, SCellBOW, unlike ItClust and scETM, does not limit model training to genes intersecting between the source and the target datasets and is independent of the source data cell type annotation. Moreover, while methods such as scETM, ItClust, and scBERT utilize numerical gene expression values directly as an input to their network architectures, SCellBOW, for the first time, encodes gene expression profiles into documents in their native format.

We compared the clustering capability of SCellBOW to popular single-cell analysis techniques as well as some recently proposed transfer learning approaches. An array of scRNA-seq datasets was used for the same, representing different sizes, cell types, batches, and diseases. As evident from the comparative analyses, SCellBOW exhibits robustness and consistency across all data and metrics. We also reported tangible benefits of transfer learning from apt source data. For example, transfer learning with a large number of tumor-adjacent normal cells from the prostate as the source and healthy prostate cells as the target adequately portrayed the cell type diversity therein. SCellBOW reliably clustered pancreatic islet-specific cells that were processed using varied single-cell technologies. The prevalence of single-cell expression profiling and the heterogeneity of PBMCs make it one of the best-studied tissue systems in humans (*Phongpreecha et al., 2020*). We isolated matched PBMCs and splenocytes from two healthy donors and two brain-dead donors (the cause of death is subarachnoid hemorrhage on aneurysm rupture). The utilization of this data presents a more intricate problem, where the cells originate from diverse biological and technical replicates, conditions, individuals, and organs. SCellBOW-based analysis of PBMCs with near-ground-truth cell type annotation confirmed its ability to adequately decipher underlying cellular heterogeneity. Our results allude to a visible improvement in scRNA-seq analysis outcomes, even with small sample sizes and multiple covariates, when contrasted with other best-practice single-cell clustering methods.

Beyond robust identification of cell type clusters, SCellBOW uses algebraic operations to analyze the cellular phenotypes that could potentially identify their contribution to patient survival outcomes. We have leveraged the power of word algebra through document embeddings to perform risk stratification of cancer subtypes based on their aggressiveness. This involves comparing the likelihood of a subtype being eliminated from the entire tumor under negative selection pressure. In addition to overall survival probability, SCellBOW assigns a risk score to discern the differences between equally aggressive subpopulations that may be hard to decipher from their survivability profiles. The *phenotype algebra* module exhibits resilience in describing the risk associated with malignant cell subpopulations arising from various types of cancer, which otherwise cannot be accomplished using gene expression data. We demonstrated a potential use case in estimating the survival risk of known molecular subtypes of three cancer types: GBM, BRCA, and mCRPC. Several examples were shown where simple algebraic operators such as '+' and '−' could derive clinically intuitive outcomes. For example, subtracting the CLA phenotype from the whole tumor resulted in an improvement in the survival risk in a GBM patient compared to MES and PRO subtypes. Upon further probing of the tumor prognosis by removal of the subtypes in combinations from the tumor, specifically, CLA and MES subtypes, which are known to be the most aggressive, we observed the highest improvement. To summarize, SCellBOW allows simulations involving multiple phenotypes. Our subsequent investigation focused on a BRCA dataset that harbors a more complex subtype structure. We observed a deviance in our results from the general perception about the relative aggressiveness of the NORMAL subtype in breast cancer. Notably, removing the NORMAL subtype from the *total tumor* was associated with the highest improvement in prognosis. This contradicts the common assumption that a NORMAL subtype is an artifact resulting from a high proportion of normal cells in the tumor specimen (*Strehl et al., 2011*). Despite indications that these tumors often do not respond to neoadjuvant chemotherapy (*Yersal and Barutca, 2014*), the clinical significance of the NORMAL subtype remains uncertain due to a limited number of studies. The NORMAL breast cancer cells are poorly characterized and heterogeneous in nature. As per the recent classification of the breast cancer subtypes, the NORMAL subtype has been identified to be a potentially aggressive molecular subtype, referred to as claudin-low tumors

(*Liu et al., 2014*). This evidence suggests that the NORMAL subtype of breast cancer is potentially an aggressive molecular subtype, which is consistent with the prediction made by SCellBOW.

In advanced metastatic prostate cancer, molecular subtypes are still poorly defined, and characterizing novel or more fine-grained subtypes with clinical implications on patient prognosis is an active field of research. As a proof-of-concept, we executed SCellBOW on the three mCRPC subtypes, namely ARAH, ARAH, and NE. We observed that eliminating the subtypes individually and in combinations (ARAH +ARAL + NE) exhibited a clinically intuitive change in prognosis using SCellBOW. We further expanded the application of *phenotype algebra* beyond its previous application solely to cancer cells by including non-malignant cells from the tumor microenvironment. We observed that eliminating cancer subtypes from the tumor decreased its aggressiveness, whereas removing the immune component increased aggressiveness, highlighting the critical role of the tumor microenvironment in modulating cancer progression. To date, mCPRC classification has largely been confined to the gradients of AR and NE activities. There is considerable scope for embracing more fine-grained subtypes to better explain clonal selection and epithelial plasticity in drug resistance in wider use cases. Convinced by the overall performance of SCellBOW, we applied *phenotype algebra* on the SCellBOW clusters obtained from the *He et al., 2021* mCRPC dataset. We observed a misalignment from the general perception that the neuroendocrine subtype features the worst prognosis. Our results pointed towards the existence of a more aggressive and dedifferentiated subpopulation in the lineage plasticity continuum. This subpopulation of de-differentiated cells in mCRPC, AR−/NE$_{low}$, exhibit greater aggressiveness and have distinct gene expression patterns that do not match those of any previously identified molecular subtypes. This novel subpopulation shares gene expression signatures with both androgen-repressed and neuroendocrine-activated genes. GSVA analysis of this hitherto unknown phenotype offered insights into its putative functional characteristics, which can be broadly defined by stemness, EMT, MET, and drug resistance. To summarize, SCellBOW clustering combined with *phenotype algebra* can provide novel insights into factettes of cancer biology. This can be used to delineate features of lineage plasticity and aid in better classification of molecular phenotypes with clinical significance. Once the subpopulations are identified, delving into the intricate mechanisms governing their resistant behavior can provide invaluable insights for designing new drugs. Further, such a tool can empower medical oncologists and oncology researchers to develop more personalized and systemic treatment regimes for cancer patients.

## Conclusion

SCellBOW is a scRNA-seq analysis tool that can be utilized to decipher molecular heterogeneity from scRNA-seq profiles and infer survival risks associated with the individual subpopulations in tumors. Our main contributions are as follows: a. We proposed SCellBOW which learns a distributed representation (fixed-length embedding per cell) of single cells. Despite being a computationally inexpensive and simple architecture, SCellBOW outperforms transformer-based approaches such as scBERT on both single-cell clustering and cancer cell risk stratification tasks. b. We also introduced the task of inferring cellular aggressiveness as a computationally tractable problem. The *phenotype algebra* module of SCellBOW performs joint representation learning of externally sourced tumor bulk RNA sequencing data (e.g. TCGA) and tumor scRNA-seq data of interest. Finally, a survival prediction model trained on bulk expression profiles is used for survival risk stratification for each of the malignant and non-malignant cell clusters derived from the tumor scRNA-seq data. We compared SCellBOW with existing best practice methods for its ability to precisely represent phenotypically divergent cell types across multiple scRNA-seq datasets, including an in-house generated human splenocyte and matched PBMC dataset. SCellBOW has proven effective in characterizing poorly defined metastatic prostate cancer. We identified a subpopulation of dedifferentiated cells in mCRPC, AR−/NElow, that exhibit greater aggressiveness and have distinct gene expression patterns that do not match those of any previously identified molecular subtypes. We could trace this back in a large-scale spatial omics atlas of 141 well-characterized metastatic prostate cancer samples at the spot resolution. In the case of breast cancer, our results indicated that the normal-like subtype, which was previously considered an artifact of high normal cell content in the tumor sample, may be among the most aggressive ones, which is concordant with recent reports. In conclusion, the robust clustering and risk-stratification capabilities of SCellBOW hold promise for tailored therapeutic interventions by identifying clinically relevant subpopulations and their impact on prognosis.

# Materials and methods

## Overview of SCellBOW components and functionalities

SCellBOW has two main applications: a. *Cell embedding and clustering.* In this work, we demonstrate single-cell representation learning using a source and target scRNA-seq data. The source data is typically any large scRNA-seq data capable of priming a neural architecture for improved representation learning on the target data of interest. Note that in this case, cells in the source or the target data need not be annotated. This transfer learning capability is meant to obtain improved fixed-length embeddings for cells in the target data by leveraging the transcriptomic patterns from existing datasets. b. *Survival risk attribution.* The *phenotype algebra* module predicts the aggressiveness associated with each malignant and non-malignant cell cluster. SCellBOW computes fixed-length embeddings jointly for the scRNA-seq target dataset (from the tumor under study) and tumor bulk RNA-seq profiles with patient-survival data (e.g. TCGA). Notably, embeddings of transcriptomes from scRNA-seq target data and bulk RNA-seq survival data have been determined concurrently by recalibrating the pre-trained model. A survival prediction model trained on the bulk RNA-seq embeddings and patient-survival data is then tested on the embeddings of the target data to make predictions about the degree of aggressiveness exhibited by the tumor variants. The steps involved in the above two tasks are depicted in *Figure 1*. The granular details involved in each of these steps can be found in the below subsections.

## Data preparation for clustering

SCellBOW follows the same preprocessing procedure for the source and target data. SCellBOW first filters the cells with less than 200 genes expressed and genes that are expressed in less than 20 cells. The thresholds may vary depending on the dimension of the dataset (*Appendix 1—table 2*). After eliminating low-quality cells and genes, SCellBOW log-normalizes the gene expression data matrix. In the first step, CPM normalization is performed, where the expression of each gene in each cell is divided by the total gene expression of the cell, multiplied by 10,000. In the second step, the normalized expression matrix is natural-log transformed after adding one as a pseudo-count. Highly variable genes are selected using the highly_variable_genes() function from the Scanpy package. SCellBOW performs a z-score scaling on the log-normalized expression matrix with the selected highly variable genes. SCellBOW can handle data in different formats, including UMI count, FPKM, and TPM. The UMI count data follow the same preprocessing procedure as above. SCellBOW skips the normalization step for TPM and FPKM since their lengths have been normalized.

## Creating a corpus from source and target data

The gene expression data matrix is analogous to the term-frequency matrix, which represents the frequency of different words in a set of documents. In the context of genomic data, a similar concept can be used to represent the expression level of a gene (words) in a set of cells (documents). Let $E \in R^{G \times C}$ be the gene expression matrix obtained from a scRNA-seq experiment, where each value $E_{g,c}$ of the matrix indicates the expression value of a gene $g \in G$ in a cell $c \in C$ obtained after Scanpy preprocessing. SCellBOW generates embeddings by taking two input datasets: a source and a target data matrix. The source dataset contains the initial weights of the neural network model, while the target data contains the cells that require clustering or malignancy potential ranking. Before generating the embeddings, SCellBOW performs feature scaling on the data matrix, rescaling each feature to a range of [0, 10] as follows.

$$E'_{g,c} = \text{int}\left(\frac{E_{g,c} - \min}{\max - \min}\right), \tag{1}$$

where scaling is applied to each entry with max = 10 and min = 0. This establishes the equivalence between term-frequency and gene expression matrix. Here, we consider that a specific gene is a word, and the expression of the gene in a cell corresponds to the number of copies of the word in a document. To scale the data matrix, we have used the MinMaxScaler() function from the scikit-learn package (*Pedregosa, 2011*). To create the corpus, SCellBOW duplicates the name of the expressed gene in a cell as many times as the gene is expressed. SCellBOW shuffles the genes in each cell to ensure a uniform distribution of genes across the cell, removing any positional bias within the dataset.

We verified that this randomization does not change the outcomes dramatically (**Appendix 2—figure 10**). The resulting gene names are treated as the words in the document. To build the vocabulary from a sequence of cells, a tag number is associated with each cell using the TaggedDocument() function in the Gensim package (**Rehurek and Sojka, 2011**). For each gene in the documents, a token is assigned using a module called tokenize with a word_tokenize() function in NLTK package (**Bird et al., 2009**) that splits gene names into tokens.

### The SCellBOW network

SCellBOW produces a low-dimensional fixed-length embedding of the single-cell transcriptome using transfer learning. The data matrix is transformed from $E' \in Z^{G \times C}$ feature space to $E'' \in R^{d \times C}$ latent feature space of d dimensions, with $d \ll G$. To generate the embeddings, SCellBOW trains a Doc2vec distributed memory model of paragraph vectors (PV-DM) model. The PV-DM model is similar to bag-of-words models in Word2vec (**Mikolov et al., 2013a**). The training corpus in Doc2vec contains a set of documents, each containing a sequence of words $W = \{w_1, w_2, .., w_T\}$ that forms a vocabulary $V$. The words within each document are treated as shared among all documents. The training objective of the model is to maximize the probability of predicting the target word $w_t$, given the context words that occur within a fixed-size window of size $n$ around $w_t$ in the whole corpus as follows

$$\text{maximize} \left( \frac{1}{T} \sum_{t=1}^{T} \log \Pr(w_t \mid w_{t-n,..,w_{t+n}}) \right). \tag{2}$$

The probability can be modeled using the hierarchical SoftMax function as follows

$$\Pr\left(w_t | w_{t-n}, .., w_{t+n}\right) = \frac{e^{y_{w_t}}}{\sum_i e^{y_i}}, \tag{3}$$

where each of $y_i$ computes the log probability of the word $w_t$ normalized by the sum of the log probabilities of all words in V. This is achieved by adjusting the weights of the hidden layer of a neural network. To build the Doc2vec model, we used the doc2vec() function in the Gensim library. The initial learning rate was set to 0.025, and the window size to 5. We chose the PV-DM training algorithm and set the embedding vector size to 300 as the default parameter. The choice of embedding vector size may be adjusted according to the size and dimensionality of the dataset.

### Fine-tuning using transfer learning

At first, SCellBOW is pre-trained with a source data matrix $E'_{src} \in Z^{G_{src} \times C_{src}}$ with $G_{src}$ genes and $C_{src}$ cells. The source data matrix sets the initial weights of the neural network model. During transfer learning, SCellBOW fine-tunes the weights learned by the pre-trained model from $E'_{src}$ using a target dataset $E'_{trg} \in Z^{G_{trg} \times C_{trg}}$ with $G_{trg}$ genes and $C_{trg}$ cells. This facilitates faster convergence of the neural network compared to starting from randomly initialized weights. The output layer of the network is a fixed-length low-dimension embedding (a vector representation) for each cell $c \in C_{trg}$. To infer the latent structure of $E_{trg}$ single-cell corpus, we used the infer_vector() function in the Gensim package to produce the dimension-reduced vectors for each cell in $C_{trg}$. This step ensures the network can map the target data into a low dimensional embedding space $R^d$; that is $E' \rightarrow E''$, where $\mathbf{E''} \in \mathbb{R}^{d \times C}$. The resulting embeddings can be used for various downstream analyses of the single-cell data, such as clustering, visualization of cell types, and *phenotype algebra*.

### Visualization of SCellBOW clusters

SCellBOW maps the cells to low-dimensional vectors in such a way that two cells with similar gene expression patterns will have the least cosine distance between their inferred vectors.

$$\text{similarity}\left(a, b\right) = \frac{a.b}{\| a \| \times \| b \|}. \tag{4}$$

After generating low-dimensional embeddings for the cells, SCellBOW identifies the groups of cells with similar gene expression patterns. To determine the clusters, SCellBOW uses the Leiden algorithm (**Traag et al., 2019**) on the embedding matrix of the target dataset. We used the leiden() function in the Scanpy package with a default resolution of 1.0. The resolution might vary in the

reported results depending on the dimension of the target dataset. To visualize the clusters within a two-dimensional space, SCellBOW uses the umap() function from the Scanpy library.

## Data preparation for *phenotype algebra*

Three independent datasets are required to perform *phenotype algebra*. Two of the three datasets are scRNA-seq datasets used for transfer learning (source and target data). SCellBOW preprocesses the source data matrix using the standard preprocessing steps. SCellBOW uses an additional bulk RNA-seq gene expression matrix (referred to as survival data). The samples are paired with survival information (e.g. vital status, days to follow-up, days to death). In the survival data, samples without follow-up time or survival status and samples with clinical information but no corresponding RNA-seq data were excluded. SCellBOW accounts for unequal cell distribution across different classes in the target data matrix. SCellBOW up samples the imbalanced target dataset by generating synthetic samples from the minority class. The value of the synthetic minority class sample is determined by interpolating between its neighboring cells from the same class. We used SMOTE (*Huang, 2015*) from the imblearn python library (*Lemaitre et al., 2016*). This confirms that the cell type proportions do not confound the output of *phenotype algebra*.

## Generating pseudo-bulk vectors for algebraic operations

SCellBOW constructs two types of pseudo-bulk vectors from the target dataset. First, it creates a pseudo-bulk reference vector by averaging the gene expression of all cells in the tumor (*total tumor*). This serves as a baseline representation of the entire tumor and acts as a reference for ranking different groups based on their relative expression profiles. Next, for each group in the dataset, SCellBOW generates a pseudo-bulk phenotype vector by averaging the gene expression of the cells within that specific group. This allows for a direct comparison between different cellular phenotypes. The phenotype vectors are constructed either based on the user-defined cell populations or SCellBOW clusters. Additionally, SCellBOW can perform algebraic operations, such as addition or subtraction, on multiple phenotype vectors to assess the combined risk of two or more cellular phenotypes. For example, in the equation $(P_a + P_b)$, $P_a$ and $P_b$ represents the phenotype vectors of two distinct cellular phenotypes. Following this, SCellBOW concatenates the bulk RNA-seq data matrix, and the reference and phenotype vectors based on common genes. We used the concatenate() function in the AnnData python package (*Virshup et al., 2021*). The resulting combined dataset is then passed to the pre-trained model, which maps it to a lower-dimensional embedding space.

## Survival risk attribution

SCellBOW infers the survival probability and predicted risk score for the user-defined tumor subtypes and SCellBOW clusters. The survival risk of the different groups is predicted by fitting a random survival forest (RSF; *Ishwaran et al., 2008*) machine learning model with SCellBOW embeddings. At first, the RSF is trained with the survival information combined with bulk RNA-seq embeddings obtained from transfer learning. During the prediction step, pseudo-bulk embeddings generated from the target dataset, serving as the test data, undergo a subtraction operation in which the cosine distance between each query embedding and the reference embedding is calculated. For example, the equations:

$$\Delta E_1 = E\left(T\right) - E\left(P_a\right), \tag{5}$$

$$\Delta E_2 = E\left(T\right) - E\left(P_a + P_b\right), \tag{6}$$

where $E\left(T\right)$ represents the embedding of the average expression for the total tumor, while $E\left(P_a\right)$ and $E\left(P_a + P_b\right)$ represent the embeddings of the phenotype vectors, individually and in combination, respectively. Here, $\Delta E_1$ and $\Delta E_2$ represent the differences between the embeddings, simulating the removal of subtypes from the whole tumor under negative selection pressure. The resulting difference is then used as input to the RSF model to infer the survival probability $S\left(t\right)$. The survival probability computes the probability of occurrence of an event beyond a given time point $t$ as follows.

$$\Pr\left[T < t\right] = \int_{-\infty}^{t} f\left(x\right) dx, \tag{7}$$

$$S(t) = 1 - \Pr[T < t] = \Pr[T > t], \tag{8}$$

where $T$ denotes the waiting time until the event occurs and $f(x)$ is the probability density function for the occurrence of an event. SCellBOW computes the survival probability using the predict_survival_function() from the scikit-survival RandomSurvivalForest python package (*Pölsterl, 2020*) with n_estimators = 1000.

In addition to survival probability, SCellBOW can estimate the relative aggressiveness of different phenotypes by assigning a risk score for distinct groups. To infer the predicted risk score, SCellBOW first trains 50 bootstrapped RSF models using 80% of the training set for each iteration. The training data is sub-sampled using different seeds for every iteration. We used the predict() function from the scikit-survival package to compute the risk score of each of the input vectors. SCellBOW derives the median of the predicted risk score for each group from the 50 bootstrapped models. The *phenotype algebra* model assigns groups with shorter survival times a lower rank by considering all possible pairs of groups in the data. The groups with a lower predicted risk score after removal from the reference pseudo-bulk are considered more aggressive, as they are associated with shorter survival times.

## Description of datasets for model evaluation

To evaluate the performance of SCellBOW, we used 15 publicly available scRNA-seq datasets and an in-house scRNA-seq dataset spanning different cell types, sizes, and diseases (*Appendix 1—table 2*). Four use cases were constructed to benchmark the clustering efficiency, each involving a pair of single-cell expression datasets. In the first use case, we used ~120,300 non-cancerous human prostate cells from *Karthaus et al., 2020* and ~28,600 healthy prostate cells from *Henry et al., 2018*. The second use case consisted of two PBMC datasets from *Zheng et al., 2017* containing approximately 68,000 cells and 2700 cells. For the third use case, we combined three independent batches of pancreatic islet cells from *Baron et al., 2016*, *Muraro et al., 2016*, and *Wang et al., 2016*, with a total of 11,181 cells as the source data and 2068 cells from *Segerstolpe et al., 2016* as the target data. In the fourth use case, we used the (*Zheng et al., 2017*) data containing approximately 68,000 cells as the source data and the in-house dataset isolated from matched spleen and PBMC samples from multiple patients as the target data.

To validate the accuracy of *phenotype algebra*, we constructed three use cases from publicly available tumor datasets comprising cells from GBM, BRCA, and mCRPC patient tumors. Each instance involved a pair of single-cell expression datasets and a bulk RNA-seq dataset paired with clinical information. For ease of reference, we stick to the following nomenclature – (a) the source data comprises the scRNA-seq dataset used for model pre-training, (b) the target data comprises cells under investigation, and (c) the survival data comprises tumor bulk RNA-seq samples. In the case of GBM, we used two scRNA-seq datasets comprising approximately 12,074 cells and 4508 cells obtained from *Neftel et al., 2019* and *Couturier et al., 2020*, respectively. In the case of BRCA, we retrieved triple-negative breast cancer scRNA-seq datasets from *Wu et al., 2020* and *Zhou et al., 2021* with approximately 24,271 cells and 545 malignant cells, respectively. In both the use cases, the bulk RNA-seq was obtained from TCGA (https://portal.gdc.cancer.gov). In the third use case, we acquired author-annotated malignant cells from *He et al., 2021*. A total of 836 malignant cells and 1334 non-malignant cells were derived from 11 patients and three metastasis sites: bone, lymph node, and liver. We retrieved 81 mCRPC bulk RNA-seq samples with paired survival from *Abida et al., 2019*. We obtained the gene expression of 141 regions of interest determined by digital spatial gene expression profiling of mCRPC from *Brady et al., 2021*. We used this data to correlate the (*He et al., 2021*) scRNA-seq gene expression with the DSP gene expression of the six phenotypic categories based on the AR- and NE- activity: AR+/NE−, $AR_{low}$/NE−, AR−/NE−, AR−/$NE_{low}$, AR+/NE+, and AR−/NE+ (Details of the datasets are described in Appendix 1, Supplementary methods 1).

## Isolation of matched PBMC and splenocyte samples

In this study, we isolated PBMC and splenocyte from four patients (two healthy donors, HD1 18 years old male and HD2 61 years old female; two brain-dead donors, BD1 57 years old female and BD2 58 years old female). PBMCs were isolated from blood after Ficoll gradient selection. Spleen tissue was mechanically dissociated and then digested with Collagenase and DNAseq. Splenocytes were finally isolated after Ficoll gradient selection. Splenocytes and PBMCs were stored in DMSO with 10% FBS in liquid nitrogen at the Biological Resource Centre for Biobanking (CHU Nantes, Hotel Dieu,

Centre de Ressources Biologiques). This biocollection was authorized in May 2013 by the French Agence de la Biomedecine (PFS13-009).

## Library preparation

For the in-house dataset, we performed CITE-seq using Hashtag Oligos (HTO) to pool samples into a single 10 X Genomics channel for scRNA-seq (*Abidi et al., 2020*). The HTO binding was performed following the specified protocol (Total-seq B). After thawing and cell washing, 1 M cells were centrifuged and resuspended in 100 μL PSE buffer (PBS/FBS/EDTA). Cells were incubated with 10 μl human FcR blocking reagent for 10 min at 4 °C. 1 μL of a Hashtag oligonucleotide (HTO) antibody (Biolegend) was then added to each sample and incubated at 4 °C for 30 min. Cells were then washed in 1 mL PSE, centrifuged at 500 × *g* for 5 min, and resuspended in 200 μL PSE. Cells were stained with 2 μL DAPI, 60 μM filtered, and viable cells were sorted (ARIA). Cells were checked for counting and viability, then pooled and counted again. Cell viability was set to <95%. Cells of HD1 spleen were lost during the washing step and thus not used during further downstream processing. Cells were then loaded on one channel of Chromium Next Controller with a 3' single-cell Next v3 kit. We followed protocol CG000185, Rev C, until the library generation stage. For the HTO library, we followed the protocol of the 10 X genomics 3' feature barcode kit (PN-1000079) to generate HTO libraries.

## Next-generation sequencing and post-sequencing quality control

We sequenced 310pM of pooled libraries on a NOVAseq6000 instrument with an S1(v1) flow-cell. The program was run as follows: Read1 29 cycles / 8 cycles (i7) /0 (i5)/Read2 93 cycles (Standard module, paired-end, two lanes). The FASTQ files were demultiplexed with CellRanger v3.0.1 (10X Genomics) and aligned on the GRCh38 human reference genome. We recovered a total of 6296 cells with CellRanger, and their gene expression matrices were loaded on R. We performed the downstream analysis of the in-house matched PBMC-splenocyte dataset in R using Seurat 4.0. For RNA and HTO quantification, we selected cell barcodes detected by both RNA and HTO. We demultiplexed the cells based on their HTO enrichment using the HTODemux() function in the Seurat R package with default parameters (*Butler et al., 2018*; *Stoeckius et al., 2018*). We subsequently eliminated doublet HTOs (maximum HTO count >1) and negative HTOs. Singlets were used for further analysis, leaving 4819 cells and 33,538 genes. We annotated each of the cell barcodes using HTO classification as PBMC and Spleen based on the origin of cells and HD1, HD2, BD1, and BD2 based on the patients (*Appendix 2—figure 4*). We performed automatic cell annotation using the Seurat-based Azimuth using human PBMC as the reference atlas (*Hao et al., 2021*; Appendix 2—note 2). We defined six major cell populations: B cells, CD4 T cells, CD8 T cells, natural killer (NK) cells, monocytes, and dendritic cells. We then manually verified the annotation based on the RNA expression of known marker genes (*Appendix 2—figure 4d*, *Appendix 1—table 4*).

## Acknowledgements

DS acknowledges the support of the ihub-Anubhuti-iiitd Foundation set up under the NM-ICPS scheme of the DST. JP, AR, PS, and SF thank the biological resource centre for biobanking (CHU Nantes, Hôtel Dieu, Centre de Ressources Biologiques (CRB), Nantes, F-44093, France (BRIF: BB-0033–00040)) and the Genomics Core Facility GenoA, member of Biogenouest and France Genomique, and to the Bioinformatics Core Facility BiRD, member of Biogenouest and Institut Français de Bioinformatique (IFB) (ANR-11-INBS-0013) for the use of their resources and their technical support.

## Additional information

### Competing interests
Gaurav Ahuja, Debarka Sengupta: is a stockholder at CareOnco BioTech Pvt. Ltd. The other authors declare that no competing interests exist.

## Funding

| Funder | Grant reference number | Author |
| --- | --- | --- |
| Department of Biotechnology | IC-12044(12)/4/2022-ICD-DBT | Debarka Sengupta |
| Science and Engineering Research Board | CRG/2022/007706 | Debarka Sengupta |

The funders had no role in study design, data collection and interpretation, or the decision to submit the work for publication.

## Author contributions

Namrata Bhattacharya, Conceptualization, Data curation, Software, Formal analysis, Validation, Investigation, Visualization, Methodology, Writing – original draft, Writing – review and editing; Anja Rockstroh, Supervision, Validation, Writing – review and editing; Sanket Suhas Deshpande, Data curation, Methodology; Sam Koshy Thomas, Anunay Yadav, Software; Chitrita Goswami, Smriti Chawla, Methodology; Pierre Solomon, Resources, Data curation; Cynthia Fourgeux, Antoine Roquilly, Resources; Gaurav Ahuja, Supervision, Methodology; Brett Hollier, Supervision; Himanshu Kumar, Validation, Investigation, Writing – original draft; Jeremie Poschmann, Resources, Software; Melanie Lehman, Supervision, Writing – original draft, Writing – review and editing; Colleen C Nelson, Supervision, Funding acquisition, Writing – review and editing; Debarka Sengupta, Supervision, Investigation, Writing – original draft

## Author ORCIDs

Namrata Bhattacharya ⓘ https://orcid.org/0000-0002-5666-2551
Gaurav Ahuja ⓘ https://orcid.org/0000-0002-2837-9361
Himanshu Kumar ⓘ https://orcid.org/0000-0001-5246-2694
Antoine Roquilly ⓘ https://orcid.org/0000-0002-1029-6242
Jeremie Poschmann ⓘ https://orcid.org/0000-0002-9613-5297
Debarka Sengupta ⓘ https://orcid.org/0000-0002-6353-5411

Reviewer #2 (Public review): https://doi.org/10.7554/eLife.98469.3.sa1
Author response https://doi.org/10.7554/eLife.98469.3.sa2

---

# Additional files

## Supplementary files

Supplementary file 1. Result of differential expression analysis for the *He et al., 2021* metastatic prostate cancer dataset. This file also includes information on custom gene sets used for ARPC and NEPC analysis.

Supplementary file 2. Result of differential expression analysis for the in-house matched PBMC and splenocyte dataset.

MDAR checklist

## Data availability

The in-house matched PBMC-splenocyte scRNA-seq expression data generated in this study is available in the GEO with accession number GSE221007. Details of the public datasets analyzed in this paper are described in *Appendix 1—table 2*. All source codes are available at GitHub (*Bhattacharya, 2025*).

The following dataset was generated:

| Author(s) | Year | Dataset title | Dataset URL | Database and Identifier |
|---|---|---|---|---|
| Bhattacharya N | 2023 | Single cell sequencing of peripheral blood mononuclear cells and spleenocytes from 4 polytraumatized patients with splenectomy | https://www.ncbi.nlm.nih.gov/geo/query/acc.cgi?acc=GSE221007 | NCBI Gene Expression Omnibus, GSE221007 |

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

# Appendix 1

**Appendix 1—table 1.** Overview of tools and benchmarking methods used in this paper.

| Tool | Version | URL | Resolution dependent | Visualization | Clustering algorithm | Reference |
|------|---------|-----|----------------------|---------------|----------------------|-----------|
| Scanpy | 1.9.1 | https://github.com/scverse/scanpy | Yes | UMAP | Louvain | *Wolf et al., 2018*; *Wolf et al., 2023* |
| ItClust | 1.2.0 | https://github.com/jianhuupenn/ItClust | No | UMAP | ItClust | *Hu et al., 2020*; *Hu, 2022* |
| Seurat | 4.1.1 | https://github.com/satijalab/seurat | Yes | UMAP | Louvian | *Butler et al., 2018*; *Butler et al., 2022* |
| DESC | 2.1.1 | https://github.com/eleozzr/desc | Yes | UMAP | Louvain | *Li et al., 2020*; *Li and Lyu, 2020* |
| scBERT | 1.0.0 | https://github.com/TencentAILabHealthcare/scBERT | Yes | UMAP | Leiden | *Yang et al., 2022*; *Tencent AI Lab Healthcare, 2022* |
| scPhere | 1.0.0 | https://github.com/klarman-cell-observatory/scPhere | Yes | UMAP | Louvain | *Ding and Regev, 2021*; *Ding, 2021* |
| scETM | 0.4.9 | https://github.com/hui2000ji/scETM | Yes | UMAP | Leiden | *Zhao et al., 2021*; *Cai, 2021* |

**Appendix 1—table 2.** Summary of datasets analyzed in this paper.

| Model | Dataset | Tissue | Technology | Data Type | Cell/sample Detected | Used in SCellBOW | Data used as | Cell filter | Gene filter | HVG |
|-------|---------|--------|------------|-----------|---------------------|------------------|--------------|-------------|-------------|-----|
| Normal Prostate | *Karthaus et al., 2020* | Human primary prostate cancer | 10 X | TPM | 120,300 | Clustering | Source | 200 | 20 | 5000 |
| | *Henry et al., 2018* | Human normal prostate | 10 X | Raw count | 28,702 | Clustering | Target | 200 | 3 | 3000 |
| PBMC | *Zheng et al., 2017* | Human PBMC | 10 X | Raw count | 68, 579 | Clustering | Source | 200 | 20 | 5000 |
| | *Zheng et al., 2017* | Human PBMC | 10 X | Raw count | 2,700 | Clustering | Target | 200 | 20 | 2000 |
| Pancreas | *Baron et al., 2016* | Human pancreas | inDrop | Raw count | 8,562 | Clustering | Source | 200 | 20 | 2000 |
| | *Muraro et al., 2016* | Human pancreas | CEL-Seq2 | Raw count | 2,042 | Clustering | Source | | | |
| | *Wang et al., 2016* | Human pancreas | SMARTer | Raw count | 430 | Clustering | Source | | | |
| | *Segerstolpe et al., 2016* | Human pancreas | Smart-Seq2 | Raw count | 2,068 | Clustering | Target | 200 | 3 | 2000 |

*Appendix 1—table 2 Continued on next page*

*Appendix 1—table 2 Continued*

| Model | Dataset | Tissue | Technology | Data Type | Cell/ sample Detected | Used in SCellBOW | Data used as | Cell filter | Gene filter | HVG |
|---|---|---|---|---|---|---|---|---|---|---|
| GBM | *Neftel et al., 2019* | Human glioblastoma | 10 X | Raw count | 12,074 | Algebra | Source | 200 | 20 | 1000 |
| | *Couturier et al., 2020* | Human glioblastoma | 10 X | Raw count | 4,508 | Algebra | Target | 200 | 3 | 1000 |
| | TCGA-GBM *Weinstein et al., 2013\** | Human glioblastoma | Bulk RNA-seq | Raw count | 613 | Algebra | Survival | | | |
| BRCA | *Wu et al., 2020* | Human breast cancer | 10 X | Raw count | 24,271 | Algebra | Source | 200 | 20 | 1000 |
| | *Zhou et al., 2021* | Human Breast cancer | Smart-seq2 | Raw count | 545 | Algebra | Target | 200 | 3 | 1000 |
| | TCGA-BRCA *Weinstein et al., 2013\** | Human Breast cancer | Bulk RNA-seq | Raw count | 1,079 | Algebra | Survival | | | |
| mCRPC | *He et al., 2021* | Human metastatic prostate cancer | Smart-Seq2 | TPM | 836 | Algebra | Target | 200 | 3 | 1000 |
| | *Abida et al., 2019* | Human metastatic prostate cancer | Bulk RNA-seq | TPM | 81 | Algebra | Survival | | | |

Data downloaded from https://www.cancer.gov/tcga.

**Appendix 1—table 3.** Summary of evaluation metric for all target datasets at resolution = 1.0 analyzed in this paper.

| | Normal Prostate | | | 3 K PBMC | | | Pancreas | | | In-house CITE-seq | | |
|---|---|---|---|---|---|---|---|---|---|---|---|---|
| | ARI | NMI | SI (Cell type) | ARI | NMI | SI (Cell type) | ARI | NMI | SI (Cell type) | ARI | NMI | SI (Cell type) |
| SCellBOW | 0.26 | 0.56 | 0.13 | 0.49 | 0.52 | 0.11 | 0.56 | 0.82 | 0.53 | 0.65 | 0.72 | 0.06 |
| Scanpy | 0.16 | 0.53 | 0.13 | 0.36 | 0.5 | 0.1 | 0.38 | 0.73 | 0.24 | 0.51 | 0.66 | 0.08 |
| Seurat | 0.15 | 0.52 | 0.21 | 0.31 | 0.48 | 0.1 | 0.52 | 0.79 | 0.53 | 0.47 | 0.66 | 0.04 |
| ItClust | 0.01 | 0.02 | −0.05 | 0.33 | 0.46 | −0.05 | 0.31 | 0.29 | −0.05 | 0.43 | 0.55 | −0.13 |
| scETM | 0.11 | 0.37 | 0.04 | 0.38 | 0.5 | 0.08 | 0.35 | 0.67 | 0.46 | 0.41 | 0.61 | −0.03 |
| scBERT | 0.18 | 0.51 | 0.22 | 0.32 | 0.47 | 0.07 | 0.35 | 0.72 | 0.35 | 0.58 | 0.70 | 0.06 |
| scPhere | 0.04 | 0.42 | 0.05 | 0.19 | 0.38 | 0.04 | 0.23 | 0.63 | 0.33 | 0.24 | 0.55 | 0.21 |
| DESC | 0.17 | 0.52 | 0.02 | 0.46 | 0.49 | −0.06 | 0.39 | 0.72 | 0.2 | 0.54 | 0.65 | 0.01 |

**Appendix 1—table 4.** Marker gene set major immune cell types.

| Major cell types | Marker genes |
|---|---|
| B cells | *CD19, CD79A, MS4A1, CD74, HLA-DRA* |
| CD4 T | *IL7R, CCR7, CD3D, CD4* |
| CD8 T | *GZMK, CD8A, CD8B, GZMB* |
| DC | *CST3, CD14, ITGAM, ITGAX* |
| MAIT | *CD3D, KLRB1, RORA, ZBTB16* |

*Appendix 1—table 4 Continued on next page*

*Appendix 1—table 4 Continued*

| Major cell types | Marker genes |
|---|---|
| Mono | CD14, S100A12 |
| NK cells | NKG7, GNLY, CD247, CCL3, GZMB, CD3D |

**Appendix 1—table 5.** Computation time across different transfer learning methods under the same hardware conditions (128 GB RAM, 16 core processor).

**Wall time (Pancreas dataset)**

| Methods | Source Model (~12 K cells) | Target Model (~2 K cells) | Total time |
|---|---|---|---|
| ItClust | 2 min 4 s | | ~2 min |
| SCellBOW (thread = 16) | 2 min 5 s | 1 min 20 s | ~3 min |
| SCellBOW (thread = 1) | 6 min 21 s | 2 min 8 s | ~8 min |
| scETM (600 epoch, thread = 16) | 22 min 38 s | 5 min 46 s | ~27 min |
| scETM (600 epoch, thread = 1) | 23 min 49 s | 5 min 58 s | ~28 min |
| scBERT | 3 hrs 33 min | 2 min | ~3 hrs |

The term 'thread' represents the number of threads used: thread = 1 indicates single-threaded execution, while thread >1 indicates multi-threaded execution.

## Supplementary methods

### 1. Datasets overview

#### 1.1 Normal prostate data

We built our pre-trained normal prostate tissue model using the large (*Karthaus et al., 2020*) dataset with approximately 120,300 cells as the source data. Cells in this study were isolated from histologically normal prostate regions of men treated for prostate cancer by radical prostatectomy. We obtained the processed TPM expression data from https://singlecell.broadinstitute.org/single_cell/study/SCP864, which was then log1p transformed, and 5000 HVGs were selected for model training. We analyzed the clustering ability of the pre-trained model on the (*Henry et al., 2018*) target dataset with approximately 28,702 cells from normal prostate specimens (young adult human prostate and prostatic urethra). The cells had been pre-annotated based on their similarity with matched single-cell transcriptomes of 9 FACS-purified cell subpopulations (Smooth Muscle (SM), Neuroendocrine epithelial (NE), Leukocytes (Leu), Luminal epithelial (LE), Hillock, Fibroblast (Fib), Endothelial (Endo), Club, and Basal epithelial (BE)). We retrieved the raw count data and cell type annotation from the GUDMAP database (https://doi.org/10.25548/W-R8CM). Raw counts were preprocessed according to '*Data preparation for clustering*' in the main methods section, and 3000 HVGs were selected for retraining. The 10 xChromium protocol was used for sequencing both source and target sample sets.

#### 1.2 Peripheral blood mononuclear cell (PBMC) data

We used a well-characterized reference 68 K PBMC dataset from *Zheng et al., 2017*, with approximately 68,579 cells obtained fresh from a healthy donor as the source dataset. We used the 3 K PBMC dataset from the same donor, consisting of 2700 cryopreserved cells, as our target data. Both datasets are hosted on the 10 x genomics website (https://www.10xgenomics.com/). We downloaded the gene-cell raw count matrix for the 'Fresh 68 k PBMCs (Donor A)' and 'Frozen PBMCs (Donor A)' as our source and target datasets, respectively. We annotated the cell types in the target dataset based on similarity with expression profiles of 11 purified subpopulations of PBMCs (CD14 +Monocyte, CD19 +B, CD34+, CD4 +Memory T, CD4 +Naive T, CD4 +T Helper2, CD4 +T Reg, CD56 +NK, CD8 +Cytotoxic T, CD8 +Naive T, Dendritic) as a reference, as described by Zheng et al. Raw counts of both data sets were preprocessed according to '*Data preparation for clustering*' in the main methods section. We selected a subset of 5000 HVGs for the 68 K PBMC and 2000 HVGs for the 3 K PBMC dataset for further analysis.

### 1.3 Pancreatic islet data

We used four single-cell RNA-seq (scRNA-seq) datasets (*Baron et al., 2016*, *Muraro et al., 2016*, *Wang et al., 2016*, and *Segerstolpe et al., 2016*) from the human pancreatic islet. The datasets are from multiple individual samples sequenced using different sequencing protocols. In the Baron dataset, the cells were sequenced using the inDrop protocol and are available on GEO: GSE84133. The Muraro dataset is available with series number GEO: GSE85241 and was sequenced using the CEL-Seq2 protocol. The Wang dataset is sequenced by the SMARTer protocol and is available under accession GEO: GSE83139. The Segerstolpe dataset was sequenced by the Smart-Seq2 protocol and was retrieved from ArrayExpress (EBI) with accession number E-MTAB-5061. We used Baron, Muraro, and Wang datasets as our source data for the pre-trained model. A total of 11,181 cells were selected. We used the Segerstolpe dataset with 2,068 cells as the target dataset. Raw counts of both data sets were preprocessed according to '*Data preparation for clustering*' in the main methods section, and 2000 HVGs were selected for further analysis. For these datasets, we used the cell labels given by the authors as ground truth. We filtered the unclassified and unclear cells from these datasets.

### 1.4 Glioblastoma data

Glioblastoma multiforme (GBM) is the most aggressive, invasive, and undifferentiated type of tumor. Glioblastomas are highly heterogeneous and have well-characterized aggressive malignant subtypes: proneural (PN), classical (CL), and mesenchymal (MES) (*Doucette et al., 2013*). We have applied *phenotype algebra* to the GBM dataset from , which consists of 4,508 malignant cells from one GBM patient (BT400). The 10 X Genomics dataset sample BT400 was downloaded from https://github.com/mbourgey/scRNA_GBM (*Bourgey, 2021*). We clustered the scRNA-seq dataset using the Leiden clustering algorithm *Traag et al., 2019* at a resolution of 1.0. We used well-known markers of cell types to perform cluster-level annotation of the dataset based on Gene Set Variation Analysis (GSVA) (*Hänzelmann et al., 2013*) scores (see Appendix 1, Supplementary methods 2.1). We have used the single-cell GBM dataset from *Neftel et al., 2019* with 12,074 cells as our source data for transfer learning. The Neftel et al. 10 X dataset was retrieved from GEO: GSE131928. The source dataset was log normalized using Scanpy, and 1000 HVGs were selected. For survival model training, we retrieved the survival and bulk RNA-seq expression data of the TCGA GBM subset with 613 samples from https://portal.gdc.cancer.gov. We used raw count expression matrices for all the datasets.

### 1.5 Breast cancer data

We executed *phenotype algebra* on PAM50-subtyped breast cancer cells from the (*Zhou et al., 2021*) study, with the raw count data available at GEO: GSE118390. The authors annotated 545 malignant cells of six triple- negative breast cancer (TNBC) patients from *Karaayvaz et al., 2018*, using the SubPred_pam50() function of the genefu R package (*Gendoo et al., 2016*). As our source data for transfer learning, we used the single-cell TNBC dataset from *Wu et al., 2020* with 24,271 cells from five patients, with the raw count data matrix downloaded from https://singlecell.broadinstitute.org/single_cell/study/SCP1106/. Raw counts were preprocessed according to '*Data preparation for clustering*' in the main methods section, and 1000 HVGs were chosen for further analysis. For training the *phenotype algebra* survival model, we used the survival and bulk RNA-seq expression data of the TCGA BRCA subset from 1079 samples retrieved from https://portal.gdc.cancer.gov.

### 1.6 Metastatic prostate cancer data

We have used a publicly available metastatic castration-resistant prostate cancer (mCRPC) scRNA-seq data set originating from multiple donors and three metastatic tumor sites (lymph node, bone, and liver) (*He et al., 2021*). In this dataset, we removed non-malignant cells based on the given cell type annotation published by the authors and focused further analysis on the 836 malignant cells derived from 11 patients. In prostate cancer, molecular subtypes are still poorly defined, and characterizing novel or more refined subtypes with clinical impact on patient prognosis is an active field of research. We employed GSVA scoring using established gene sets describing the molecular traits to classify 836 malignant cells. We annotated the tumor cells into one of the three categories: ARAH, NE, and ARAL (see Appendix 1, Supplementary methods 2.2). We used the gene expression data and metadata from the publication. We retrieved survival and bulk RNA-seq expression data for 81 mCRPC patients provided by *Abida et al., 2019* for training the survival model. We retrieved the dataset from https://www.cbioportal.org/ (prad_su2c_2019). We have used the Karthaus et al.

pre-trained model as our source model for transfer learning. All samples available were individually normalized with transcripts per million (TPM).

## 2. Molecular characterization of the cancer datasets

### 2.1 Annotation of Couturier et al. GBM dataset

For molecular characterization of the Couturier et al. datasets, we performed gene set enrichment analysis using GSVA scoring with default settings on the raw expression data. The gene sets comprise known marker genes (*Verhaak et al., 2010*) for the Proneural (PRO), Classical (CLA), and Mesenchymal (MES) subtypes, as outlined in *Appendix 1—table 6*.

**Appendix 1—table 6.** Gene set for molecular subtypes of Glioblastoma.

| Subtype | Marker genes |
| --- | --- |
| Proneural | *DLL3, BCAN, OLIG2, NCAM1, NKX2-2, ASCL1, PDGFRA* |
| Classical | *EGFR, CDKN2A, RB1, CDK4, CCDN2* |
| Mesenchymal | *CHI3L1, CD44, VIM, RELB, TRADD, PDPN, YKL40, MET, NF1, TNFRSF1A* |

We applied SCellBOW clustering on the dataset at a resolution of 1.0 and obtained 10 clusters. We performed a cluster-level subtype annotation based on the GSVA scores of the cells. For each cluster, we have assigned a molecular subtype based on the prevalent subtype (percentage of the cells) in that cluster. Clusters CL3, CL4, CL5, and CL6 have a higher percentage of PRO than MES and CLA and have thus been annotated as PRO. Similarly, clusters CL2, CL7, and CL8 were assigned MES, and clusters CL0, CL1, and CL9 were assigned CLA.

### 2.2 Annotation of He et al. mCRPC dataset

For phenotypic characterization of the metastatic cells in the mCRPC target data, we performed GSVA scoring with default parameters on the preprocessed log1p(TPM) expression data. We used published marker gene sets (*Appendix 1—table 7*) for 'androgen receptor-positive prostate cancer' (ARPC) and 'neuroendocrine prostate cancer' (NEPC) from *Labrecque et al., 2019*, two well-described types of advanced prostate cancer.

**Appendix 1—table 7.** *Labrecque et al., 2019* gene sets for molecular subtypes of mCRPC.

| Subtype | Marker genes |
| --- | --- |
| NEPC | *CHGA, SYP, ACTL6B, SNAP25, INSM1, ASCL1, CHRNB2, SRRM4* |
| ARPC | *AR, NKX3-1, KLK3, CHRNA2, SLC45A3, NAP1L2, S100A14, TRGC1, TARP* |

To obtain a high-level subgrouping, we performed K-means (*Lloyd, 1982*) clustering on the ARPC and NEPC signature scores from GSVA using K-means++initialization (*Appendix 2—figure 8a and b*). Based on the elbow method, the optimal number of clusters was determined to be 3. For further molecular characterization, we performed differential gene expression analysis on the obtained clusters using the *rank_genes_groups* function in Scanpy python library and, based on the results, annotated three subgroups of (i) 'Androgen Receptor Activity High' (ARAH), (ii) 'Androgen Receptor Activity Low' (ARAL), and (iii) 'neuroendocrine-like' (NE) cells. (*Supplementary file 1*). The ARAH cluster was characterized by high expression of AR-activated (*TRGC1, TRGC2, KLK3, KLK2, GNMT, NKX3-1, NAP1L2, TMPRSS2, SLC45A3, C1orf116*) and NE-repressed genes (AR, SLC25A37, RGS10); while showing low expression of NE-activated genes (*SRRM4, SYP, SYT11, GNA01, KCNB2, GPX2, ETV5, INSM1, DNMT1, TRIM9, SNAP25*). The NE cluster exhibited high expression of NE-activated genes (*CHGA, SCG3, PROX1, ASCL1, SNAP25, TRIM9, INSM1, GPX2, EZH2, DNMT1, KCNB2, SYP, SYT11, SRRM4*); as wells as low expression of NE-repressed (*GATA2, RGS10, SLC25A37, MAPKAPK3, RIPK2, RGS10*) and AR activated genes (*ABCC4, CENPN, SLC45A3, NAP1L2, C1orf116, GNMT, FKBP5, PMEPA1, TRGC1*). The ARAL cluster had attenuated AR expression with measurable but low expression of some AR-activated (*GNMT, NAP1L2, TRGC1, TRGC2*) and NE-activated genes (*SYP, SYT11, GNAO1, GPX2, KCNB2, INSM1, PROX1, SCG3, ASCL1, CHGA*). Executing *phenotype algebra* on these broad subgroups resulted in a risk prediction matching the expectation and current knowledge.

## 3. Gene set enrichment analysis on the mCRPC He et al. dataset

For further characterization of the metastatic data set, not biased by previous knowledge or expectations, we performed SCellBOW clustering on the preprocessed in Scanpy log1p function on TPM expression data of the 836 malignant cells using resolution 0.8 and obtained eight clusters. We GSVA-scored as above, but against an extensive collection of ~16,500 gene sets describing a wide variety of cellular processes. We utilized gene sets from the Molecular Signatures Database (MSigDB.v.7.1:http://www.gsea-msigdb.org/gsea/downloads_archive.jsp H: hallmark gene sets, C2: curated gene sets and C5: ontology gene sets) in combination with in-house curated custom gene signatures (*Supplementary file 1*). Performing differential gene set analysis across the SCellBOW clusters using rank_genes_groups function in Scanpy python library. with default settings using a Wilcoxon Rank Sum test and Bonferroni p-value adjustment revealed activated and repressed pathways that were employed to describe the biology of clusters.

## Appendix 2

### Supplementary Notes

#### Note 1: Comparison of SCellBOW clustering efficacy with additional methods

For benchmarking, we further compared the performance of SCellBOW with two recently published tools- scPhere (**Ding and Regev, 2021**) and scBERT (**Yang et al., 2022**). scBERT handles scRNA-seq data by leveraging contextualized embeddings built upon BERT (**Devlin et al., 2018**), which capture the context and relationships between genes in a cell's transcriptional profile. Its primary emphasis lies in cell annotation, particularly focusing on cell type classification. On the other hand, scPhere is a deep generative model that embeds cells into geometric spaces, enabling a unique way of representing cells within specialized geometric dimensions compared to traditional Euclidean spaces. For comparing the clustering ability, we used the Normal prostate, pancreatic islet, 3 K PBMC, and our in-house PBMC-spleen datasets for our benchmarking (see SCellBOW effectively captures latent space of single-cell phenotypes in the main text).

We computed the adjusted Rand index (ARI) for a resolution ranging from 0.2 to 2.0 in 0.2 intervals and found that SCellBOW consistently outperformed scBERT and scPhere across all resolutions (**Appendix 2—figure 3a–d**). Then, for the same range of resolutions, we computed the normalized mutual information (NMI) (**Appendix 2—figure 3e**) and observed that scPhere consistently exhibited poorer performance compared to SCellBOW and scBERT across all datasets. SCellBOW outperformed scBERT across all public datasets. In terms of the Silhouette Index (SI) for cell types, SCellBOW demonstrated superior performance over scPhere and scBERT in the 3 K PBMC and Pancreas datasets (**Appendix 2—figure 3f**). However, in the in-house CITE-seq dataset, scPhere displayed improved results, while scBERT exhibited enhanced performance in the normal prostate dataset.

#### Note 2: Processing of in-house PBMC-spleen dataset in SCellBOW

##### Dataset

This CITE-seq dataset includes 4,819 cells and 33,538 genes from the spleen and matching PBMC of 4 polytraumatized patients with splenectomy hemostasis, generated by us using 10 X.

##### Cell and gene filtering criteria

(1) eliminated cells with gene counts <200 using filter_cells() function from Scanpy Python package; (2) eliminated genes if the number of cells expressing this gene is <3 filter_genes() function.

##### Data processing

(1) gene expression levels for each cell were normalized using the normalize_total() function in Scanpy with target_sum = 10,000; (2) normalized gene expression was then transformed using log(1+x) transformation with natural logarithm using log1p function; (3) top 3,000 highly variable genes were selected using the highly_variable_genes() function in Scanpy; (4) the expression is further standardized to a z-score using scale function, and the standardized gene expression values were used as input for SCellBOW. After the above filtering and data processing, 4,785 cells and 3,000 highly variable genes remained for downstream analysis.

##### Clustering and visualization using SCellBOW

All 4,785 cells were clustered by SCellBOW with default parameters using the leiden() function. We used the umap to visualize the 2D plot of the clusters in SCellBOW.

##### Cell type labeling of CITE-seq using Azimuth

Cell type annotation was performed using Seurat-based Azimuth (**Hao et al., 2021**). The molecular reference atlas for Human-PBMC was used for annotation from https://azimuth.hubmapconsortium.org/. We eliminated the cells with a cell type count <5. There were 4,817 remaining cells.

#### Note 3: Erythroid cells in the PBMC-spleen dataset

In our analysis, we observed that cells identified as 'Eryth' by Azimuth are scattered across all the clusters. Since erythroid cells were removed during the CITE-seq experiments, erythroid cells are not part of PBMCs in this dataset. Thus, we assumed that if erythrocytes are present, it is most

likely due to contamination. Since erythroid cells form a majority of cluster CL6 amidst T cells, they are putative ambient RNA contaminations misguiding the Azimuth annotation (*Appendix 2—figure 4e*). We further performed differential gene expression analysis based on cell types using the Scanpy rank_genes_groups function on the PBMC-spleen dataset. We observed that they have mitochondrial genes as highly differentially expressed genes. This can be attributed to the process of erythropoiesis in cells (*Gonzalez-Ibanez et al., 2020*). However, given the spatial distribution of cells classified as 'Eryth' across multiple clusters combined with high mitochondrial gene content in the cells, we assume that 'Eryth' cells are misclassified/unclassified cells (*Appendix 2—figure 4g*).

## Note 4: Dependency of *phenotype algebra* on SCellBOW embeddings

The concept of *phenotype algebra* is new, and there is no algorithm that is directly comparable to *phenotype algebra* on the specific task of sub-clonal survival risk (aggressiveness) stratification. However, *phenotype algebra* is built upon fixed-length embeddings generated in SCellBOW. To assess the dependency of *phenotype algebra* on these embeddings, we examined the possibility of achieving risk inference directly from raw gene expression data for known molecular subtypes of specific cancers (BRCA, GBM, mCRPC). In the case of GBM, we observed that the gene expression model failed to correctly position CLA, MES, and (CLA +MES) in terms of their aggressiveness (*Appendix 2—figure 6i*). For BRCA, while it effectively positioned LUMA and HER2, it incorrectly categorized LUMB as the most aggressive subtype, despite LUMB typically having a favorable prognosis (*Appendix 2—figure 6j*). Similarly, in mCRPC, the gene expression-based model identified the NE subtype as the least aggressive, contradicting established literature, which recognizes NE as the most aggressive subtype (*Appendix 2—figure 6k*). These results demonstrated that using the survival modeling techniques directly on gene expression data, as opposed to SCellBOW embeddings, failed to accomplish accurate risk stratification.

To illustrate the effectiveness of fixed-length embeddings obtained from SCellBOW for *phenotype algebra,* we compared them to other tools such as scETM (*Hao et al., 2021*), scPhere (*Ding and Regev, 2021*), and scBERT (*Yang et al., 2022*). Unlike SCellBOW, which is an unsupervised transfer learning-based model, the limitation of scETM and scBERT is the dependency on the cell type labeling of the source dataset. Due to the lack of source data annotation in Neftel et al. GBM dataset, we used the Scanpy clusters as the source annotation. scPhere is a deep generative model that embeds cells into low-dimensional space and doesn't need any source data. In the case of scPhere and scBERT, we have used embedding size = 300 to match SCellBOW. However, in the case of scETM, we adhered to the default embedding size of 50, since it does not explicitly allow the tuning of this parameter.

In the case of GBM and mCRPC, we observed that all the tools (scETM, scPhere, and scBERT) failed to detect the relative position of the subtypes in the combined state ((MES +CLA) for GBM and (ARAH +ARAL + NE) for mCRPC) (*Appendix 2—figure 7*). While scETM and scPhere correctly identified the least aggressive subtype in GBM as PRO; it failed to position CLA and MES accurately (*Appendix 2—figure 7a–c*). Similarly, in BRCA, we observed that scETM and scPhere could identify LUMA as the least aggressive subtype, however, they failed to detect the relative position of the remaining subtypes (*Appendix 2—figure 7d–f*). scBERT outcomes didn't align with the established aggressiveness order both in GBM and BRCA. In mCRPC, the known aggressiveness order is NE > ARAL > ARAH. scETM, scPhere, and scBERT disagreed with the known relationship among these subtypes (*Appendix 2—figure 7g–i*). scETM, scBERT, and scPhere struggle to differentiate cancer clones based on their aggressiveness and their impact on disease prognosis.

## Note 5: The importance of choosing Doc2vec as our base model

Single-cell RNA-seq profiles can relate cellular states and mRNA expression relationships, revealing gene expression programs corresponding to tumorigenesis. Identifying the groups of cells with similar phenotypic profiles is a critical step in cellular heterogeneity dissection. Natural language is a valuable analogy for cell signals where cells are analogous to sentences, and genes are analogous to words. Any downstream analysis of the scRNA-seq requires data to be represented in fewer dimensions. Embeddings are fixed-length vector representations. Using the cell sentences analogy, SCellBOW creates single-cell embedding based on the natural language processing (NLP) - Doc2vec language model (*Mikolov et al., 2013b*). We have tested the performance of SCellBOW compared to Scanpy and Doc2Vec (*Le and Mikolov, 2014*) on Segerstolpe et al. pancreas data. As source data

for SCellBOW pre-trained model, we have taken Baron et al. Both datasets were retrieved from ItClust (https://github.com/jianhuupenn/ItClust; *Hu, 2022*).

In *Appendix 2—figure 9*, we observed that the majority of cells of similar cell types are more localized in the 2-dimensional space compared to other Scanpy and Doc2vec. After SCellBOW, Doc2vec was able to group cells with similar signatures closer to each other (*Appendix 2— figure 9g–i*). Thus, embedding generated by the Doc2vec-based approaches preserves spatial segregation across the clusters. We observed the highest overall quality scores (ARI, NMI, and cell type SI) in SCellBOW (see *Appendix 2—figure 9j*). The SCellBOW score being higher than the Doc2Vec embedding and Scanpy embeddings suggest that although the Doc2vec model has good performance compared to Scanpy, transfer learning using the Doc2vec model yields higher clustering accuracy.

## Note 6: Extended analysis of differentially expressed genes in spleen against PBMC

Spleen is the largest secondary lymphoid organ in the body and orchestrates a wide range of immunological functions. The physical organization of the spleen allows it to filter blood-borne pathogens and antigens. Differential expression analysis between immune cells originating from the spleen and peripheral blood exhibited spleen-specific elevated expression of dissociation-associated genes (*JUNB*, *FOSB*, *JUND*, *JUN*) and stress response heat shock genes (*HSP*, *HSP90AB1*, *HSPD1*, *DNAJB1*) (*Appendix 2—figure 4h and i* and *Supplementary file 2*). Upregulation of dissociation and stress response-associated gene families can be attributed to the additional dissociation step during sample preparation, which is exclusive to the splenocytes (*Denisenko et al., 2020*; *Van den Brink et al., 2017*). Furthermore, B cell differentiation-related genes such as *VPREB3*, *MS4A1*, and *CD83* were found upregulated among splenocytes as compared to PBMC, suggesting the dominance of various stages of development, differentiation, and maturation of B cells in the spleen. Notably, the spleen has different structures, such as primary follicles and the germinal center within the white pulp where various stages of B cells reside. We also spotted spleen-specific upregulation of nuclear receptors (*NR4A1*, *NR4A2*) and B cell receptors (BCR) (*BANK1*, *CD79A*, *CD79B*), indicating that B cells are inherently different between circulation and the spleen and likely undergo additional maturation steps in the spleen. We also observed spleen-specific expression of B cell inhibitory receptors (BCIR) (*CD22*, *CD72*, *LY9*), suggesting that multipotent progenitors reside in the spleen. These multipotent progenitors of the spleen stem cells could be differentiating to myeloid and erythroid lineage cells in addition to differentiation of the lymphoid lineage cells (primarily to B cells and to some extent to T cells) (*Tsubata, 2016*). Overall, the BCR and BCIR are two important receptors that play complementary roles in regulating B cell activation and tolerance. While the BCR initiates signaling cascades that lead to B cell activation, the BCIR limits these signals and helps maintain B cell tolerance.

B cells are capable of developing a wide range of effector functions, including antibody secretion via differentiating to plasma cells, antigen processing and presentation to T helper cells, cytokine production, and generation of immunological memory. The mature spleen plays an important role in B cell development and antigen-dependent maturation because it is the site of terminal differentiation for developing B cells that arrive from the bone marrow known as Transitional B cells of type 1 (T1), and these cells develop to transitional B cells of type 2 (T2) and reside to the primary follicles of the spleen (*Nolte et al., 2004*). Naive B cells enter the spleen from the circulation, where they differentiate into either B effectors that differentiate into antibody-secreting plasma cells or B memory cells (*Tsai et al., 2019*). The prominent role of the spleen in B cell differentiation and activation inspired us to delve deeper into compositional and phenotypic differences associated with B cell subtypes across peripheral blood and spleen. Based on annotation, as expected, we noted enrichment of B effectors in the spleen (*Appendix 2—figure 4j*). SCellBOW embedding could spatially segregate B effectors, B naive cells, and plasmablasts, although effectors and naive cells were not split into two distinct clusters (*Appendix 2—figure 4k*).

Further analysis of B cell subtypes showed higher expression of the dissociation genes and stress response of genes in spleen B effector and B naive cells as compared to their peripheral blood counterpart (*Appendix 2—figure 4l and m*). B naive has higher expression of actin genes (*ACTB*, *ACTG1*) as compared to B effectors, suggesting the naive B cells are more dynamic for differentiation to any of B cell subtypes, such as the effector or memory (*Liu et al., 2012*). Notably, the mutation in these genes results in the development of large B-cell lymphoma. In the case of

the B effector, we observed that the expression of *CD27*, *HLADQA1*, *CD83*, *PRMT1*, and *CD69* are significantly higher in the mature effector B cells (*Infantino et al., 2017*; *Zandvoort et al., 2001*). Collectively, genes upregulated in the B cells from the spleen were found to be involved in cell-cell adhesion, activation of the B cell receptor signaling pathway, and differentiation of B cells during hematopoiesis and our analysis very well corroborates with known B cell development biology under physiological conditions.

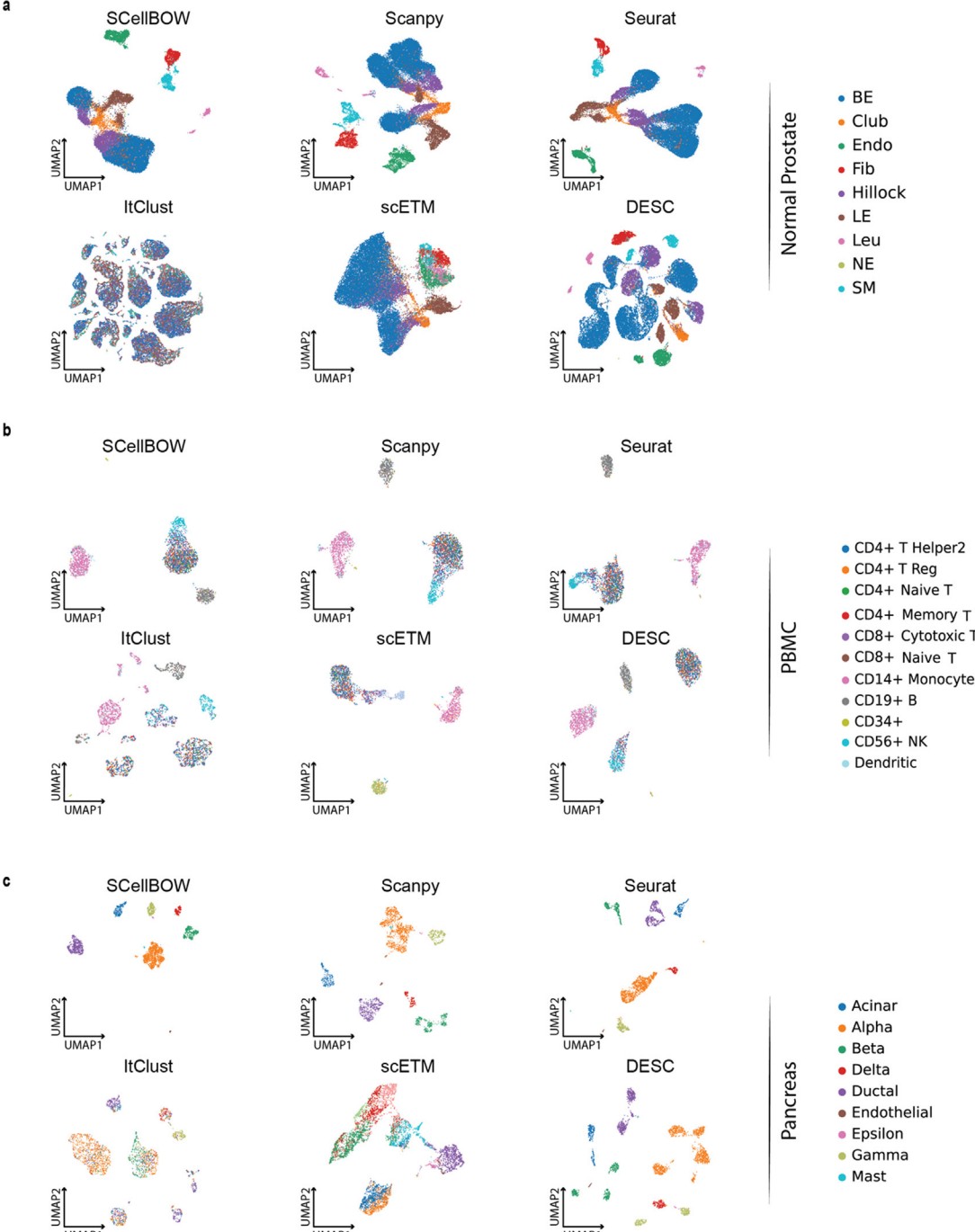

**Appendix 2—figure 1.** Cell embeddings visualization. (**a-c**) The UMAP plots showing embedding of SCellBOW compared to different existing methods benchmarked on normal prostate (**a**), peripheral blood mononuclear cells (PBMC) (**b**), and pancreas datasets (**c**). The coordinates of all the plots are colored by true cell types.

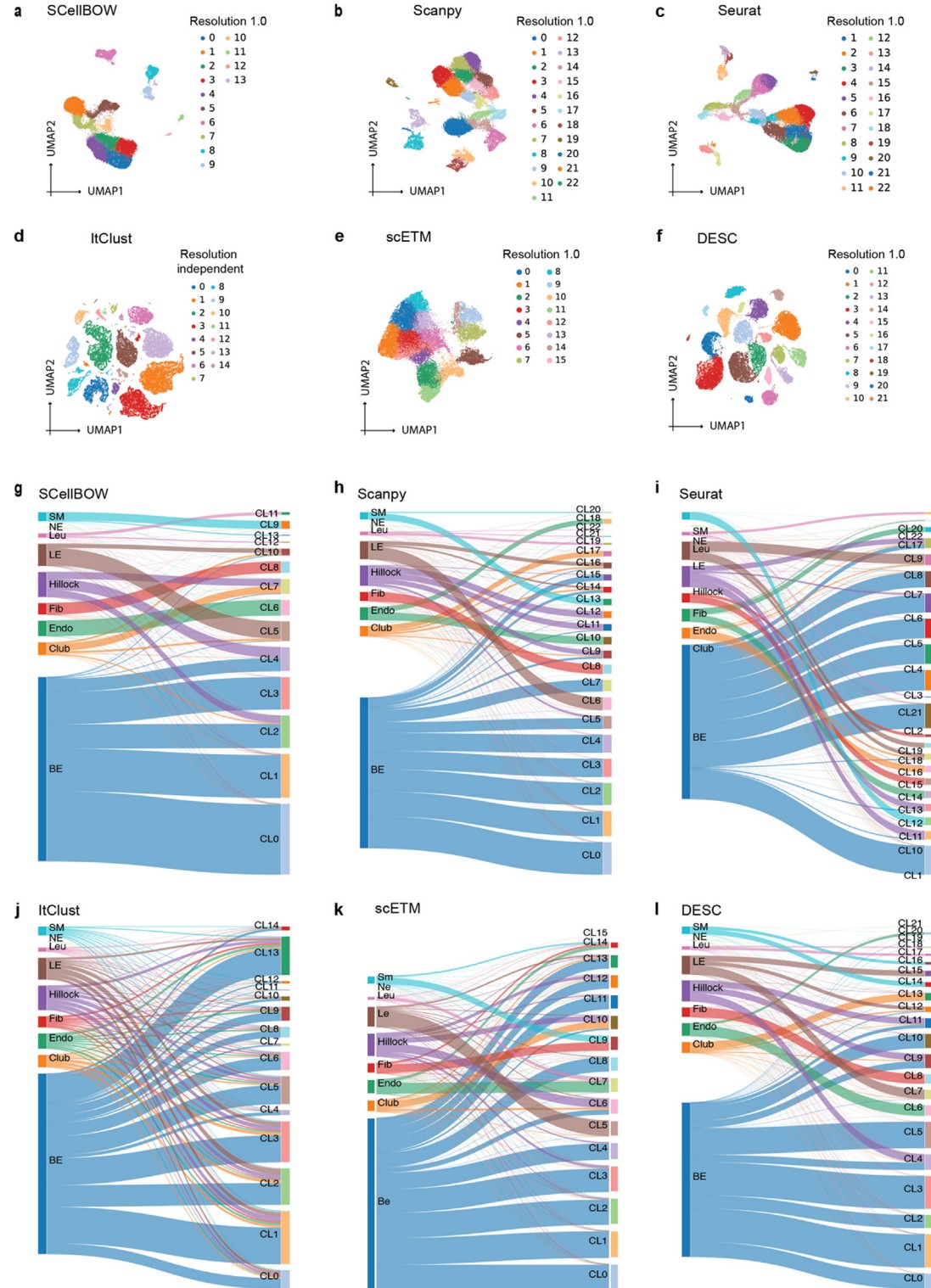

**Appendix 2—figure 2.** Cell embedding visualization of the normal prostate scRNA-seq dataset. (**a–f**) The UMAP plots showing embedding of SCellBOW, Scanpy, Seurat, ItClust, ProjectR, and DESC on normal prostate. The coordinates of all the plots are colored by clusters. (**g–l**) Alluvial plots showing the mapping of clusters resulting from the benchmarking tools onto the true cell types from Henry et al. normal prostate dataset. CL is used as an abbreviation for cluster.

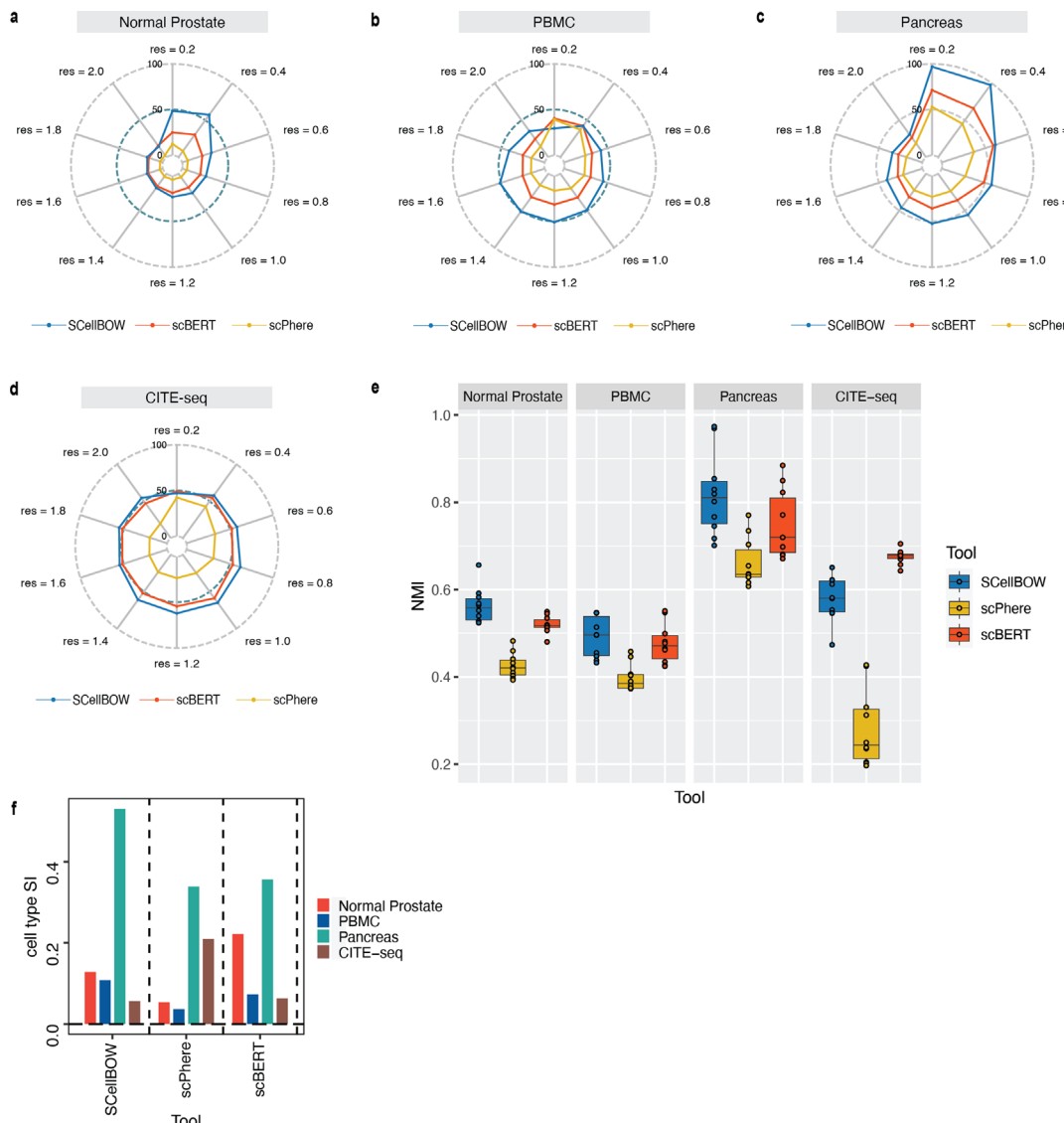

**Appendix 2—figure 3.** Extended performance evaluation of SCellBOW with scBERT and scPhere. (**a–d**) Radial plot for the percentage of contribution of different methods towards ARI for various resolutions ranging from 0.2 to 2.0 for normal prostate (**a**), PBMC (**b**), pancreas (**c**), and CITE-seq (**d**) datasets. (**e**) Box plot for the NMI of different methods across different resolutions ranging from 0.2 to 2.0 in steps of 0.2. (**f**) Bar plot for the cell type silhouette index (SI) for different methods. The default resolution was set to 1.0.

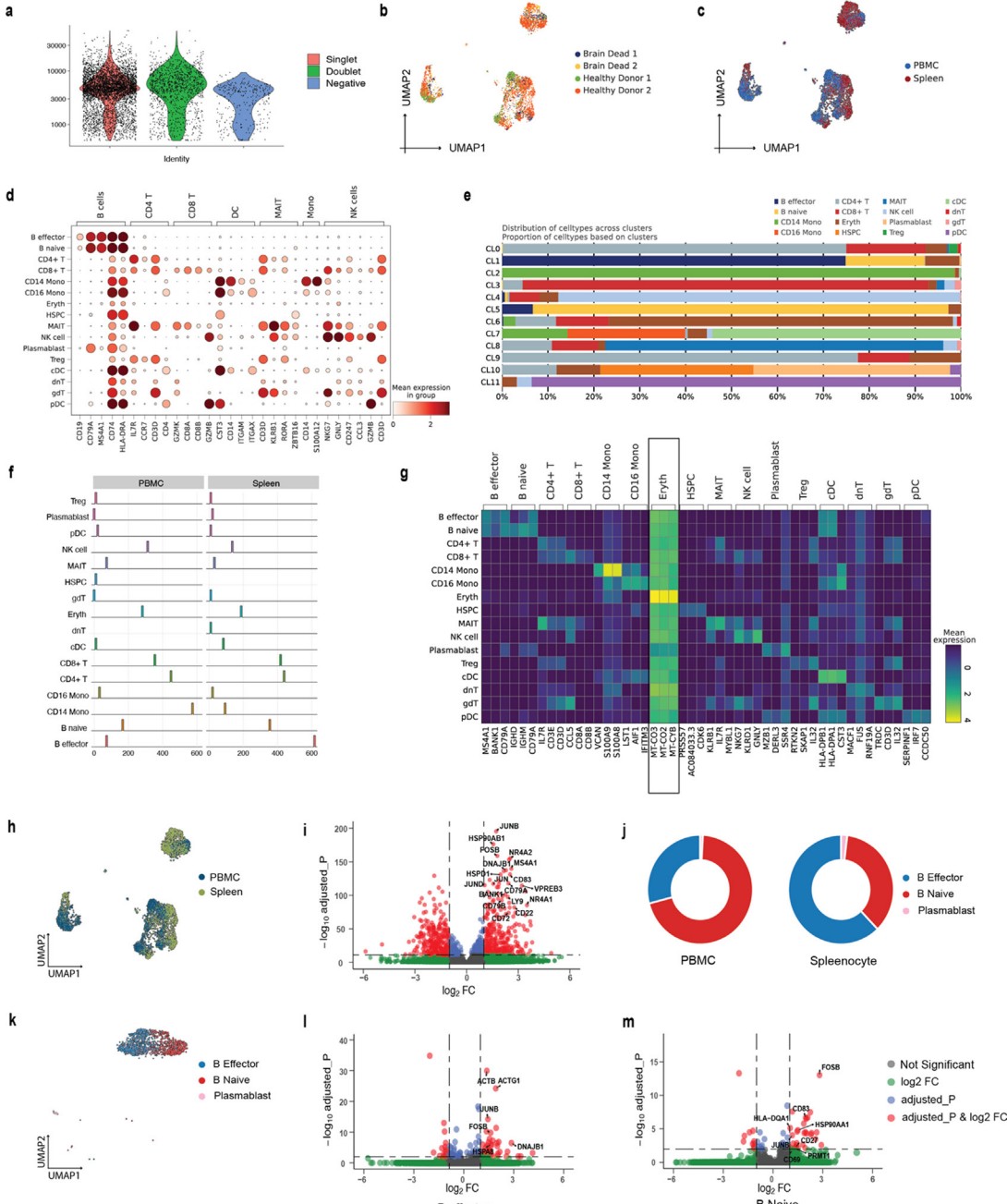

**Appendix 2—figure 4.** Extended analysis of in-house CITE-seq dataset. (**a**) Violin plot showing the distribution of UMIs for singlets, doublets, and negative cells. (**b**) The UMAP plots showing embedding of SCellBOW colored by donors. (**c**) The UMAP plots showing embedding of SCellBOW colored by tissue of origin. (**d**) Dot plot to check the expression of marker genes of PBMC per cell type identified by Azimuth. (**e**) Bar plot showing the proportion of annotated cell types across different clusters of SCellBOW. (**f**) Compositional difference in proportion annotated cell types in PBMC vs. splenocytes. (**g**) Heatmap for annotated cell type-wise differentially expressed genes in each cell type. (**h**) UMAP plots of the cells colored by their tissue source. (**i**) Volcano plot showing the differential genes (red dots) in the spleen and PBMC for B cells (p-value < 0.05, False discovery rate (FDR)<0.01). (**j**) Donut plot showing the compositional difference in the proportion of B cell subtypes (B naive, B effector, and plasmablast) in PBMC and spleen. (**k**) UMAP plot showing the embedding of SCellBOW colored by B cell subtypes. (**l, m**) Volcano plot showing the differential genes (red dots) in the spleen and PBMC for B effector (**l**) and B naive cells (**m**) (p-value <0.05, FDR < 0.01).

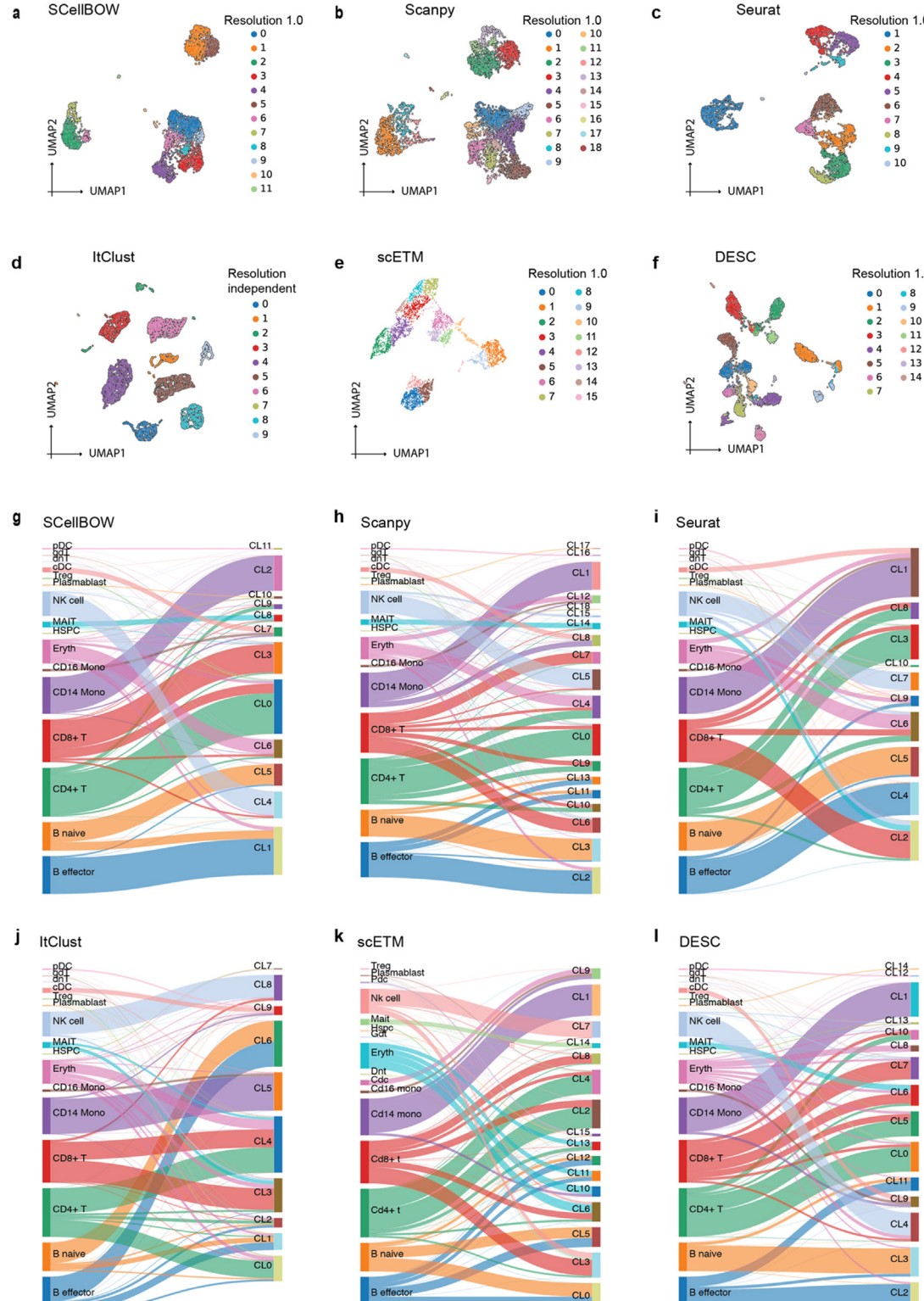

**Appendix 2—figure 5.** Cell embedding visualization of the in-house CITE-seq scRNA-seq dataset. (**a–f**) The UMAP plots showing embedding of SCellBOW, Scanpy, Seurat, ItClust, ProjectR, and DESC on the PBMC-spleen dataset. The coordinates of all the plots are colored by clusters. (**g–l**) The alluvial plots showing the mapping of clusters resulting from the benchmarking tools onto the cell types identified by Azimuth.

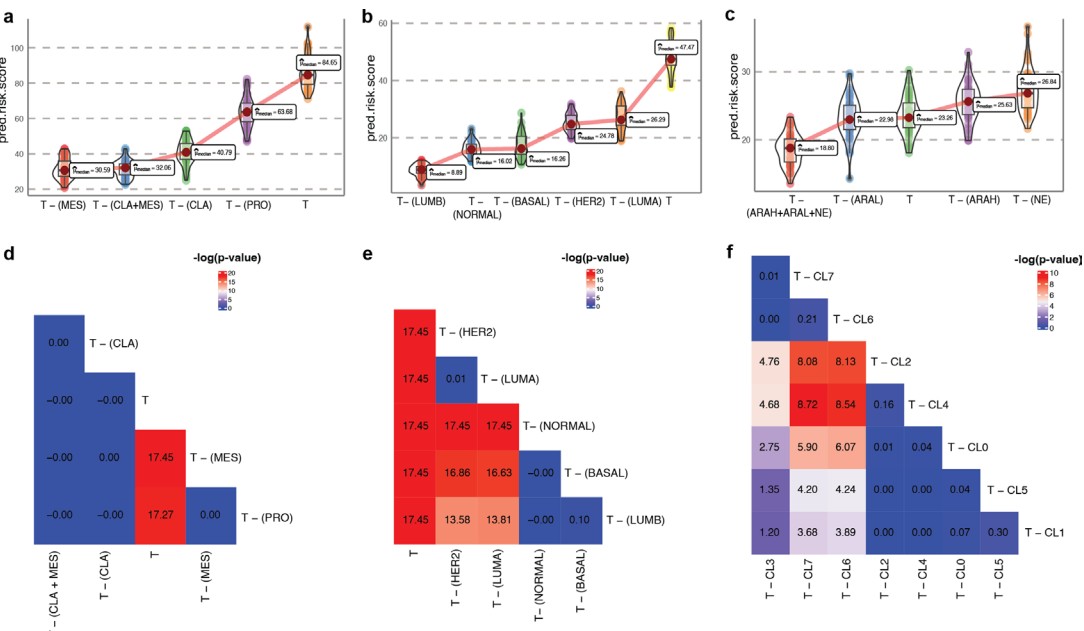

**Appendix 2—figure 6.** Survival risk inference using *phenotype algebra* on raw gene expression data. (**a–c**) *Phenotype algebra*-based risk scores using gene expression profile of GBM molecular subtypes (**a**), PAM50 molecular subtypes of BRCA (**b**), three high-level categories of mCRPC (**c**). The *total tumor* is denoted by *T*. (**d-f**) Heatmap for -log$_{10}$(p-value) of the predicted risk scores for GBM subtype (**d**), BRCA subtype (**e**), and mCRPC clusters (**f**), and using Wilcoxon unpaired one-sided test.

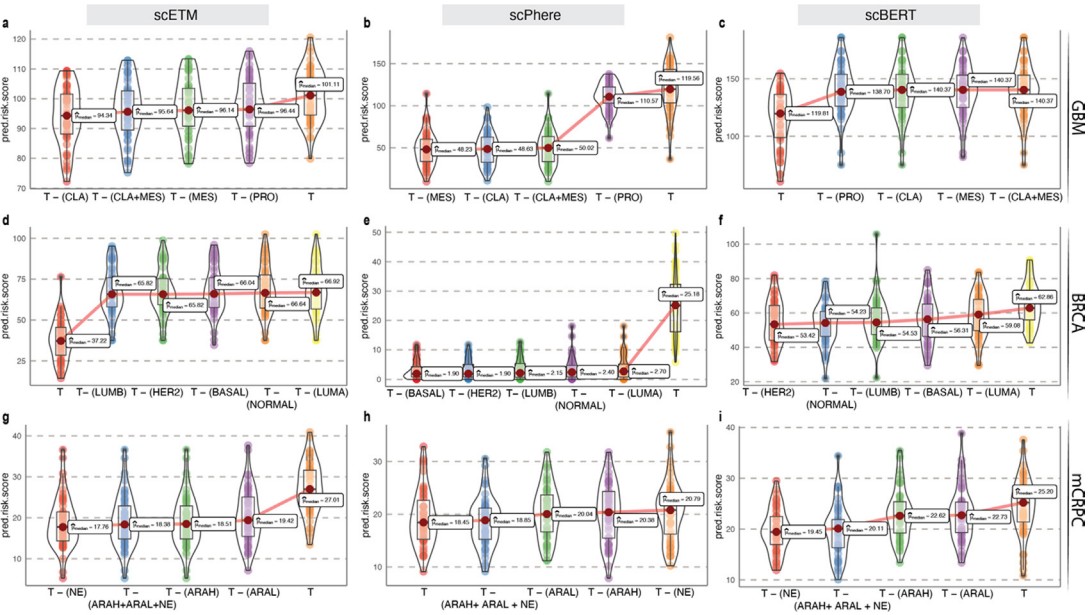

**Appendix 2—figure 7.** Survival risk inference using fixed-length embeddings from scETM, scPhere, and scBERT. (**a–c**) *Phenotype algebra*-based risk scores of GBM molecular subtypes using fixed-length embeddings from scETM (**a**), scPhere (**b**), and scBERT (**c**). The *total tumor* is denoted by *T*. (**d-f**) *Phenotype algebra*-based risk scores of PAM50 molecular subtypes of BRCA using fixed-length embeddings from scETM (**a**), scPhere (**b**), and scBERT (**c**). (**g–i**) *Phenotype algebra*-based risk scores of three high-level categories of mCRPC using fixed-length embeddings from scETM (**a**), scPhere (**b**), and scBERT (**c**).

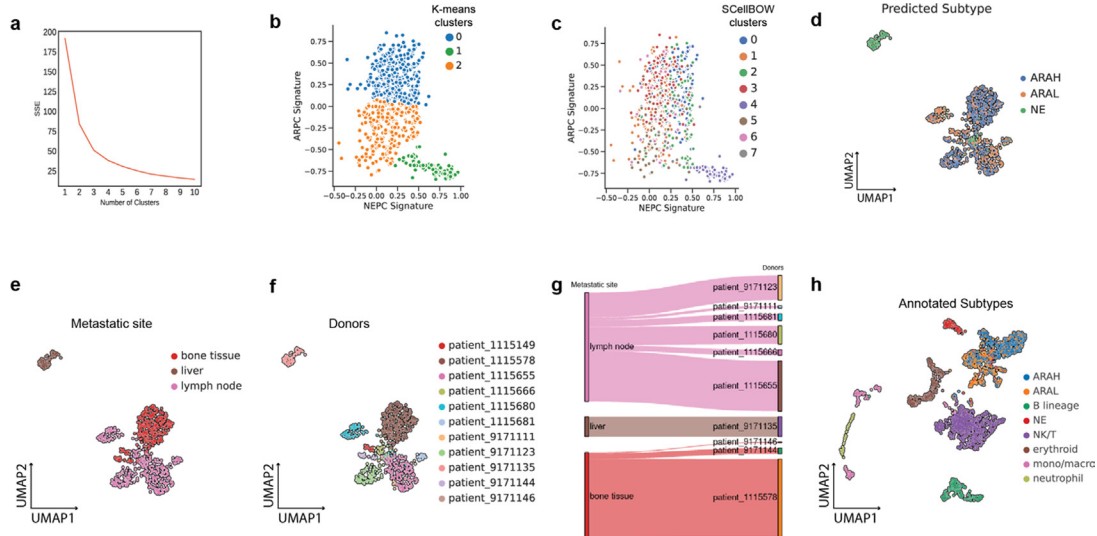

**Appendix 2—figure 8.** Extended analysis of *He et al., 2021* single-cell mCPRC dataset. (**a**) Elbow plot for selecting the best K for K-means clustering. (**b, c**) Scatter plot of GSVA scores of ARPC and NEPC gene sets colored by the K-means clusters (E) and SCellBOW clusters (F). (**d**) UMAP plot visualizing the high-level ARAH, ARAL, and NEPC categories on SCellBOW embeddings. (**e, f**) The UMAP plots showing the embedding of SCellBOW colored by metastasis site (**a**) and donors (**b**). (**g**) Alluvial plot to visualize tumor metastasis site of the donors. (**h**) UMAP plot visualizing SCellBOW embeddings of tumor microenvironment cells (malignant +non-malignant) in the He et al. dataset based on author annotations.

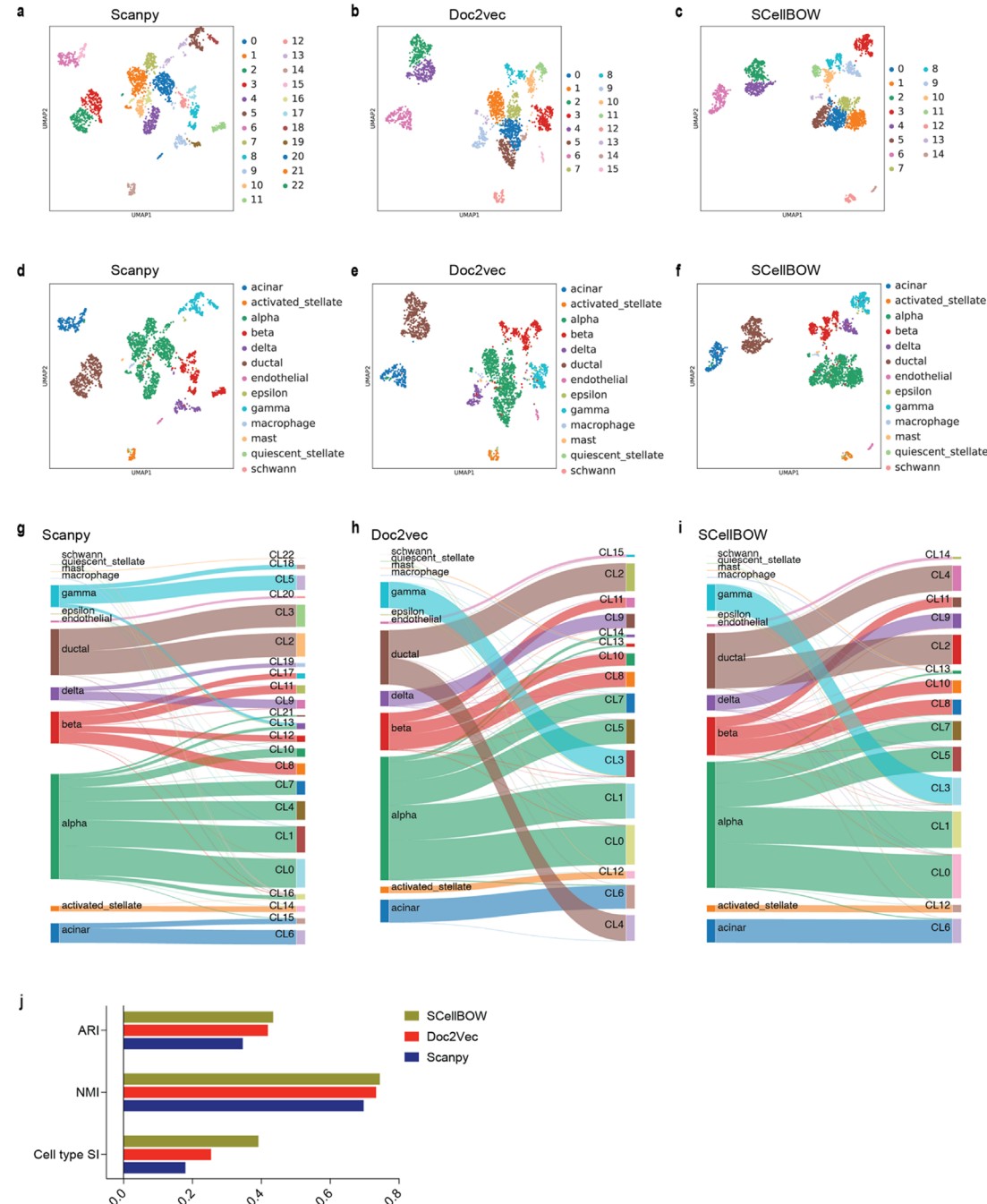

**Appendix 2—figure 9.** Assessing the quality of clustering using transfer learning on BOW models using the pancreas dataset. (**a–c**) UMAP plot of Scanpy embedding (**a**), Doc2Vec embedding (**b**), and SCellBOW embedding (**c**) of pancreas dataset using Leiden clustering at resolution 1.0. (**d-f**) UMAP plot of Scanpy embedding (**d**), Doc2Vec embedding (**e**), and SCellBOW embedding (**f**) of pancreas dataset colored with their annotated cell types (**g–i**) Alluvial plot for cell types against Leiden clusters for Scanpy (**g**) Doc2vec (**h**) SCellBOW (**i**). (**j**) Barplot for ARI, NMI, cluster purity, Silhouette index (cell type and cluster).

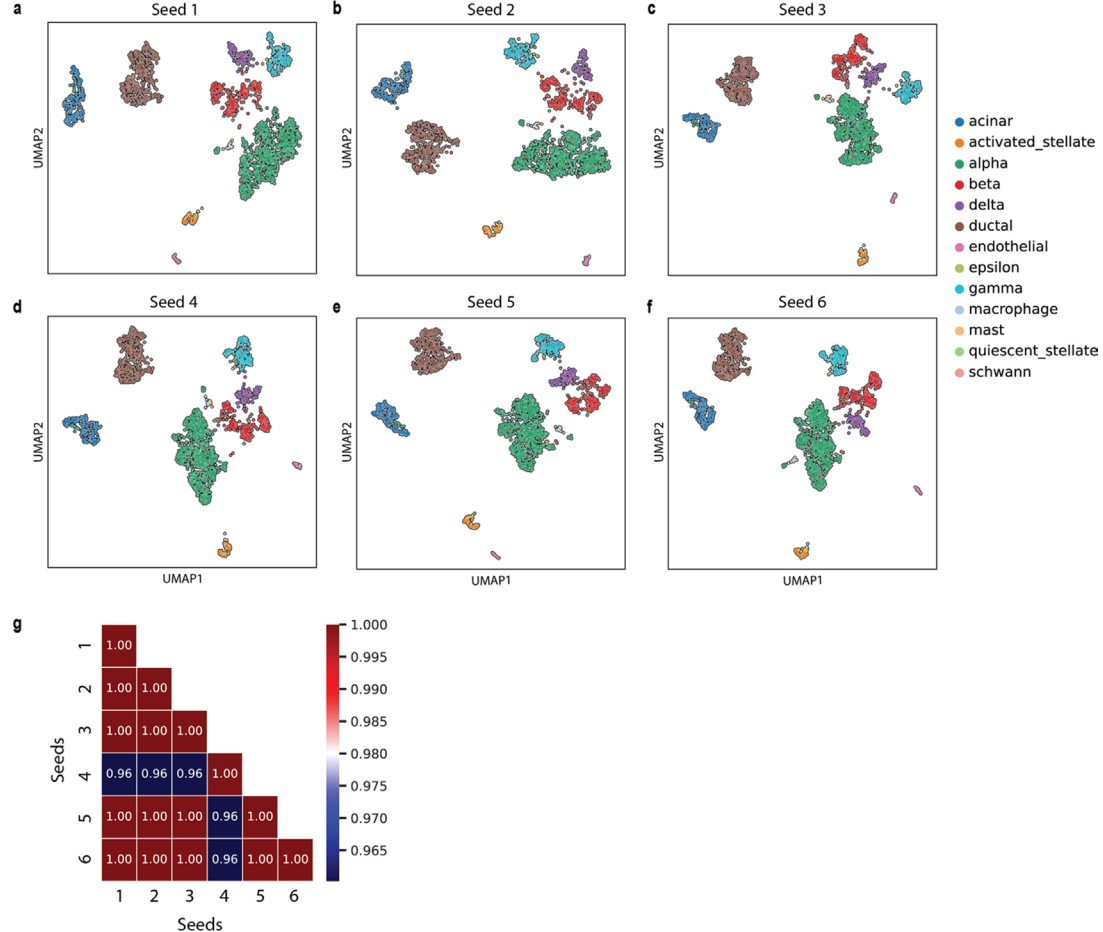

**Appendix 2—figure 10.** Assessing the effect of random seed on the quality of cell sentence generation using the pancreas dataset. (**a–f**) The UMAP plots showing the embedding of SCellBOW generated using cell sentences with different random seeds ranging from 2 to 25. The colors of the cells in the UMAP plots indicate clusters. (**g**) Heatmap showing the ARI between each pair of clustering outcomes with distinct seeds.

