## [Editor Report · eLife Assessment]

This manuscript presents an **important** contribution to the field of single-cell transcriptomic analysis in cancer by introducing a novel computational framework-SCellBOW-which applies embedding techniques from natural language processing to model phenotypic heterogeneity in tumors. The revised version includes new validation experiments and significant clarifications that provide **convincing** evidence for the method's utility. The authors have benchmarked SCellBOW across diverse datasets, including glioblastoma, breast, and metastatic prostate cancer, and have demonstrated its superior performance compared to existing state-of-the-art methods.

---

## [Referee Report · Reviewer #2 (Public review)]

Summary:

The authors developed a novel tool, SCellBOW, to perform cell clustering and infer survival risks on individual cancer cell clusters from the single cell RNA seq dataset. The key ideas/techniques used in the tool include transfer learning, bag of words (BOW), and phenotype algebra which is similar to word algebra from natural language processing (NLP). Comparisons with existing methods demonstrated that SCellBOW provides superior clustering results and exhibits robust performance across a wide range of datasets. Importantly, a distinguishing feature of SCellBOW compared to other tools is its ability to assign risk scores to specific cancer cell clusters. Using SCellBOW, the authors identified a new group of prostate cancer cells characterized by a highly aggressive and dedifferentiated phenotype.

Strengths:

The application of natural language processing (NLP) to single-cell RNA sequencing (scRNA-seq) datasets is both smart and insightful. Encoding gene expression levels as word frequencies is a creative way to apply text analysis techniques to biological data. When combined with transfer learning, this approach enhances our ability to describe the heterogeneity of different cells, offering a novel method for understanding the biological behavior of individual cells and surpassing the capabilities of existing cell clustering methods. Moreover, the ability of the package to predict risk, particularly within cancer datasets, significantly expands the potential applications.

Weaknesses:

Given the promising nature of this tool, it would be beneficial for the authors to test the risk-stratification functionality on other types of tumors with high heterogeneity, such as liver and pancreatic cancers, which currently lack clinically relevant and well-recognized stratification methods. Additionally, it would be worthwhile to investigate how the tool could be applied to spatial transcriptomics by analyzing cell embeddings from different layers within these tissues.

---

## [Author Response]

The following is the authors’ response to the original reviews

**Reviewer #1 (Public Review):**
This review evaluates the SCellBOW framework, which applies phenotype algebra to obtain vectors from cancer subclusters or user-defined subclusters.Strengths:SCellBOW employs an innovative application of NLP-inspired techniques to analyze scRNA-seq data, facilitating the identification and visualization of phenotypically divergent cell subpopulations. The framework demonstrates robustness in accurately representing various cell types across multiple datasets, highlighting its versatility and utility in different biological contexts. By simulating the impact of specific malignant subpopulations on disease prognosis, SCellBOW provides valuable insights into the relative risk and aggressiveness of cancer subpopulations, which is crucial for personalized therapeutic strategies. The identification of a previously unknown and aggressive AR−/NElow subpopulation in metastatic prostate cancer underscores the potential of SCellBOW in uncovering clinically significant findings.Major concerns:The reliance on bulk RNA-seq data as a reference raises concerns about potentially misleading results due to the presence of RNA expression from immune cells in the TME. It is unclear if SCellBOW adequately addresses this issue, which could affect the accuracy of the cancer subcluster vectors.

We appreciate the reviewer's concerns. To address the concern about potentially misleading results due to the TME when using bulk RNA-seq data as a reference:

a. We account for systematic biases between the single-cell and bulk transcriptomics readouts by creating pseudo-bulk profiles for single-cell clusters, enabling more accurate comparisons [Section Materials and methods, Data preparation for phenotype algebra].

b. We encode expressions into word vectors and co-embed them together. By doing this, we mitigate any possibility of systematic differences in the embedding. It is imperative that we subject both single-cell and bulk data through the same treatments because otherwise, it will be difficult to perform algebraic operations on them [Section Materials and methods, Generating vectors for phenotype algebra].

c. In our new analysis of the tumor microenvironment, we have shown that SCellBOW effectively differentiates between malignant and non-malignant cells, confirming that it is not biased by the immune cell composition in the bulk RNA-seq data [Section SCellBOW facilitates survival-risk attribution of tumor subpopulations, Fig. 5g-h].

The method of extracting vectors in phenotype algebra appears to be a straightforward subtraction operation. This simplicity might limit its efficiency in excluding associations with phenotypes from specific subpopulations, potentially leading to inaccurate interpretations of the data.

Thanks for this excellent query. Vector algebra operations are not done in the gene expression space (i.e., gene expression vectors associated with tumor samples), rather we process the single cell and bulk expression profiles through multiple steps (pseudo-bulk vector generation for single cell clusters, mapping gene expression values to word frequencies as better understood by the Doc2vec neural networks etc.) to ensure their embeddings are consistent and capture intricate phenotypic information. We have demonstrated this through rigorous validation of the clusters yielded on various types of healthy and diseased samples. Furthermore, we have demonstrated the consistency of the vector algebra operations on known cancer subtypes in breast cancer, glioblastoma, and prostate cancer. We have clarified this further in text. [Section Materials and methods, ‘Generating vectors for phenotype algebra’, ‘Survival risk attribution’].

The review would benefit from additional validation studies to assess the effectiveness of SCellBOW in distinguishing between cancerous and non-cancerous signals, particularly in heterogeneous tumor environments.

We thank the reviewer for advising this additional validation. While our study primarily focused on signals from malignant cells, we have now considered the impact of the tumor microenvironment. We observed that the predicted risk score increases when the immune component is subtracted from the tumor, suggesting that tumor aggressiveness increases in the absence of immune components. Importantly, the aggressiveness ranking of tumor subtypes (NE > ARAL > ARAH) remained consistent, confirming that SCellBOW effectively preserves subtype-specific risk stratification [Section SCellBOW facilitates survival-risk attribution of tumor subpopulations, Fig. 5g-h].

Further clarification on how SCellBOW handles mixed-cell populations within bulk RNA-seq data would strengthen the evaluation of its applicability and reliability in diverse research settings.

We really appreciate the reviewer’s observation. We clarify that rather than relying on absolute gene expression values, SCellBOW maps bulk RNA-seq data into an embedding space, where we extract the latent representation of the tumor. This process effectively masks the influence of mixed-cell populations, reducing biases introduced by immune or stromal components. Furthermore, *phenotype algebra* operates within this embedding space by comparing cosine similarities between latent representations of bulk and pseudo-bulk datasets, rather than using direct gene expression values. This allows SCellBOW to capture biologically meaningful relationships and infer tumor-specific signals effectively, even in the presence of heterogeneous cell populations. Our benchmarking across diverse cancer types confirms its effectiveness [Section Results, ‘SCellBOW enables pseudo-grading of metastatic prostate cancer tumor microenvironment’, ‘Unsupervised risk-stratification of metastatic prostate cancer clusters using SCellBOW’].

**Reviewer #2 (Public Review):**
The authors developed a novel tool, SCellBOW, to perform cell clustering and infer survival risks on individual cancer cell clusters from the single-cell RNA seq dataset. The key ideas/techniques used in the tool include transfer learning, bag of words (BOW), and phenotype algebra which is similar to word algebra from natural language processing (NLP). Comparisons with existing methods demonstrated that SCellBOW provides superior clustering results and exhibits robust performance across a wide range of datasets. Importantly, a distinguishing feature of SCellBOW compared to other tools is its ability to assign risk scores to specific cancer cell clusters. Using SCellBOW, the authors identified a new group of prostate cancer cells characterized by a highly aggressive and dedifferentiated phenotype.Strengths:The application of natural language processing (NLP) to single-cell RNA sequencing (scRNA-seq) datasets is both smart and insightful. Encoding gene expression levels as word frequencies is a creative way to apply text analysis techniques to biological data. When combined with transfer learning, this approach enhances our ability to describe the heterogeneity of different cells, offering a novel method for understanding the biological behavior of individual cells and surpassing the capabilities of existing cell clustering methods. Moreover, the ability of the package to predict risk, particularly within cancer datasets, significantly expands the potential applications.Major concerns:Given the promising nature of this tool, it would be beneficial for the authors to test the risk-stratification functionality on other types of tumors with high heterogeneity, such as liver and pancreatic cancers, which currently lack clinically relevant and well-recognized stratification methods. Additionally, it would be worthwhile to investigate how the tool could be applied to spatial transcriptomics by analyzing cell embeddings from different layers within these tissue

(1) We completely agree with the reviewer’s view. Our selection of glioblastoma and breast cancer for this study was primarily driven by the focus on extensively studied and well-defined cancer types. To demonstrate the effectiveness of our model, we tested it on advanced prostate cancer, which currently lacks clinically relevant and well-recognized stratification methods. This application to metastatic prostate cancer serves as a proof of concept, illustrating our model's potential to provide valuable insights into cancer types where established stratification approaches are limited or absent.

(2) Regarding the application of our tool to spatial transcriptomics, we have already analyzed data from Digital Spatial Profiling (DSP). The article is already quite complex and involved, and we are afraid the inclusion of spatial transcriptomics may amount to a significant extension of the method. To this end, although we will discuss the future possibilities, we will skip the method validity check on spatial transcriptomics data.

**Reviewer #2 (Recommendations For The Authors):**
(1) "SCellBOW adapts the popular document-embedding model Doc2vec for single-cell latent representation learning, which can be used for downstream analysis...": Using only simple gene frequency might overlook the dependent relationships between genes, potentially compromising the biological significance. This could be discussed further.

This is an excellent point raised by the reviewer. We acknowledge that using only simple gene frequency may overlook dependent relationships between genes, potentially compromising biological significance. To address this, we have now compared SCellBOW on the specific task of *phenotype algebra* and demonstrated its effectiveness in capturing meaningful biological relationships which is overlooked by simple gene frequency. We have now added the results of this comparison and showed that gene expression data alone couldn't cut it for accurate risk stratification [Section Overall discussion, Supplementary Note 7, Supplementary Fig. 8i-k].

(2) "While existing methods effectively reveal the subpopulations, they are insufficient in associating malignant risk with specific cellular subpopulations identified from scRNA-seq data....": Perhaps I missed it in the methods section, but how does SCellBOW compare to simply performing pseudobulk analysis on separate cell clusters, treating them as bulk RNA-seq, and then associating the signatures with disease prognosis?

This is an insightful point, and we appreciate the opportunity to clarify it.

(1) While pseudobulk analysis on separate cell clusters, followed by associating their signatures with disease prognosis, is a common approach, SCellBOW achieves this without requiring a priori knowledge of prognostic biomarkers to determine whether a subpopulation is aggressive.

(2) Moreover, pseudobulk analysis aggregates gene expression across cells, which can potentially mask intra-cluster heterogeneity, thereby obscuring important signatures associated with disease prognosis. In contrast, the latent representation in SCellBOW captures the semantic meaning of disease aggressiveness, allowing for a more nuanced and biologically meaningful risk assessment.

(3) "The proposed approach, SCellBOW, can effectively capture the heterogeneity and risk associated with each phenotype, enabling the identification and assessment of malignant cell subtypes in tumors directly from scRNA-seq gene expression profiles, thereby eliminating the need for marker genes...": Have the author compared the resulting group with well-known markers and do they overlap?

We appreciate this thoughtful question. While SCellBOW does not rely on predefined marker genes for clustering or risk stratification, we have systematically evaluated whether the resulting subpopulations align with well-known markers. To assess this, we compared SCellBOW-derived clusters with established marker-based annotations across multiple datasets. We observed a significant overlap between SCellBOW clusters and canonical marker-defined cell types in various cancers, including GBM, BRCA, and mCRPC.

(4) "We constructed three use cases leveraging publicly available scRNA-seq datasets...": The three training and testing datasets are all from healthy tissue. How about in tumor tissue? i.e., Could SCellBOW also identify better cell clusters in tumor datasets?

We appreciate the reviewer’s inquiry. For benchmarking and method validation, we primarily selected normal tissue datasets as they are heavily annotated and well-characterized. Our goal was to extensively evaluate SCellBOW across different clustering metrics, including ARI, NMI, and SI, which required datasets with reliable ground truth. Tumor datasets, in contrast, often lack confirmatory ground truth, making direct benchmarking more challenging. However, to assess SCellBOW’s applicability in tumor settings, we performed downstream analyses on tumor scRNA-seq datasets using phenotype algebra. Our results demonstrate that SCellBOW effectively identifies distinct cell clusters, including malignant and non-malignant populations, reinforcing its applicability in tumor settings [Section Results, ‘Unsupervised risk-stratification of metastatic prostate cancer clusters using SCellBOW’].

Minor issues:(1) Labels of subplots within the manu/figure should be revised to ensure correct order (missing Figures 3a-d, 4b before 4a, etc).

We thank the reviewer for pointing this out. We have corrected the figure labels and ensured that all subplots follow the correct order, aligning with the manuscript.

(2) "reaffirmed the clinically known aggressiveness order, i.e., CLA >-MES >-PRO, where CLA succeeds the rest of the subtypes in aggressiveness48 (Figures 4c, d)...": "Fig. 4c, d" should be "Fig. 4e, f". Also please put Figure 4a before 4b. Overall the order of Figure 4 needs to be revised to match the order in the manu. Similar to Figure 6.

We have corrected the figure reference to Fig. 4e, f and revised the order of Figure 4 to maintain consistency with the manuscript.

(3) "Our results showed that SCellBOW learned latent representation of single-cells accurately captures the 'semantics' associated with cellular phenotypes and allows algebraic operations such as'+' and'-'." Figure 5f (SCellBOW performances on mCRPC) should also be cited here since Supplementary Figure 6 contains three datasets (GBM, BRCA, mCRPC) while in Figure 4 only GBM and BRCA were shown?

We thank the reviewer for this suggestion. We have now cited Figure 5f in this section to ensure that all datasets, including mCRPC, are appropriately referenced.

(4) Under the subheading "SCellBOW facilitates survival-risk attribution of tumor subpopulations", the lines start with "We refer to this as phenotype algebra. We utilized this ability to find an association between the embedding vectors, representing total tumor - a specific malignant cell cluster with tumor aggressiveness..." could be reduced a little bit especially the re-intro of phenotype algebra since the author has already discussed previously (under "overview of SCellBOW").

We appreciate the feedback and have condensed this section to avoid redundancy while maintaining clarity in connecting phenotype algebra to survival-risk attribution.

(5) "Most CD4+ T cells map to CL0 and CL9 (here, CL is used as an abbreviation for cluster) (Figure 3f)..." "(here, CL is used as an abbreviation for cluster)" this note could be moved forward to SF2 since CL is first introduced in SF2.

We thank the reviewer for the suggestion. We have moved the definition of CL (cluster) to Supplementary Figure 2 (SF2), where it is first introduced, for improved clarity.